# *Fast&Fair*: Training Acceleration and Bias Mitigation for GNNs

**O. Deniz Kose**                                                                 *okose@uci.edu*
*Department of Electrical Engineering and Computer Science*
*University of California Irvine*

**Yanning Shen**\*                                                                *yannings@uci.edu*
*Department of Electrical Engineering and Computer Science*
*University of California Irvine*

**Reviewed on OpenReview:** *https://openreview.net/forum?id=nOk4XEB7Ke*

## Abstract

Graph neural networks (GNNs) have been demonstrated to achieve state-of-the-art performance for a number of graph-based learning tasks, which leads to a rise in their employment in various domains. However, it has been shown that GNNs may inherit and even amplify bias within training data, which leads to unfair results towards certain sensitive groups. Meanwhile, training of GNNs introduces additional challenges, such as slow convergence and possible instability. Faced with these limitations, this work proposes FairNorm, a unified normalization-based framework that reduces the bias in GNN-based learning while also providing provably faster convergence. Specifically, FairNorm presents individual normalization operators over different sensitive groups and introduces fairness regularizers on the learnable parameters of normalization layers to reduce the bias in GNNs. The design of the proposed regularizers is built upon analyses that illuminate the sources of bias in graph-based learning. Experiments on node classification over real-world networks demonstrate the efficiency of the proposed scheme in improving fairness in terms of statistical parity and equal opportunity compared to fairness-aware baselines. In addition, it is empirically shown that the proposed framework leads to faster convergence compared to the naive baseline where no normalization is employed.

## 1 Introduction

Graphs are powerful tools for modeling complex systems and the relations within them. Hence, they are widely employed to represent various real-world systems, such as gene networks, traffic networks, and social networks to name a few. Such expressiveness has led to rising attention towards learning over graphs, and it has been shown that graph neural networks (GNNs) achieve the state-of-the-art for several tasks over graphs (Gori et al., 2005; Scarselli et al., 2008; Hamilton et al., 2017a; Kipf & Welling, 2017; Veličković et al., 2018; Xu et al., 2018b). GNNs create node representations by repeatedly aggregating information from the neighbors, which can be employed on ensuing tasks such as traffic forecasting (Opolka et al., 2019), crime forecasting (Jin et al., 2020), and recommender systems (Ying et al., 2018).

Machine learning (ML) models have been widely used in various domains to make critical decisions. Therefore, it is essential to prevent discriminatory behavior in these models towards under-represented groups. However, it has been demonstrated that ML models propagate the potential bias within the training data (Dwork et al., 2012; Beutel et al., 2017) and lead to discriminatory results in ensuing applications. Particular to GNNs, it has been shown that in addition to propagating the already existing bias, GNN-based

---

\*Corresponding author.

learning may even amplify it due to the utilization of biased graph topologies (Dai & Wang, 2021). This well-motivates the studies in fairness-aware GNN-based learning.

Normalization operations shift and scale the hidden representations created in deep neural networks (DNNs) in order to accelerate the optimization process in training (Ioffe & Szegedy, 2015; Ulyanov et al., 2016; Ba et al., 2016; Salimans & Kingma, 2016; Xiong et al., 2020; Miyato et al., 2018; Wu & He, 2018; Santurkar et al., 2018). While the other aspects of GNN-based learning are theoretically investigated, such as generalization (Scarselli et al., 2018; Xu et al., 2019b) and expressiveness (Xu et al., 2018a; Loukas, 2020; Ying et al., 2021), the optimization of GNNs is analytically an under-explored area. Practically, training GNNs generally has a slow convergence rate and is accompanied by instability issues (Xu et al., 2018a). Inspired by this, Cai et al. (2021) investigates the effect of a shift operation on a simple GNN-based learning environment and proposes a normalization framework that is suitable for GNNs. The proposed framework in Cai et al. (2021), GraphNorm, is demonstrated to be more effective in improving convergence speed over graphs compared to previously presented normalization strategies in other domains.

It has been shown in Balunovic et al. (2021) that the distributional discrepancy between the representations among different sensitive groups is one of the leading factors to bias in general ML algorithms. For GNNs, fairness analyses have also shown that the distributions of the representations of different sensitive groups are key factors that affect the resulting bias (Li et al., 2020; Kose & Shen, 2022). Note that normalization layers learn the parameters that manage the sample mean and variance of these hidden representations. Hence, the normalization can be readily applied to manipulate the related statistics to reduce the bias, while also improving the convergence. Motivated by this, this study proposes a unified GNN-based learning framework, FairNorm, that provides a faster convergence through the employment of a normalization layer, while also mitigating the bias with novel fairness-aware regularizers on the learnable parameters in the introduced normalization layer. Overall, the contributions of the present work can be summarized as follows:

**c1)** We propose a framework that can reduce bias while providing a higher convergence speed for a GNN-based learning environment. To the best of our knowledge, FairNorm is the first attempt to improve fairness and convergence speed in a unified framework.

**c2)** The effect of the proposed shift operations on convergence rate is investigated in a simple GNN-based learning framework. It is analytically shown that the proposed shift operations can improve the convergence rate for node classification compared to the case where no shift is employed.

**c3)** Fairness-aware regularizers are introduced on the trainable parameters of the normalization layers. The design of the regularizers is based on theoretical understanding regarding the sources of bias in GNNs.

**c4)** Empirical results are obtained over real-world networks in terms of utility and fairness metrics for node classification. It is demonstrated that compared to fairness-aware baselines, FairNorm leads to an improvement in fairness metrics while providing comparable utility. Meanwhile, it is empirically shown that FairNorm enhances the convergence speed with respect to the no-normalization baseline.

## 2 Related Work

**Fairness-aware learning over graphs:** Rahman et al. (2019) serves as a seminal work for random walk-based fairness-aware learning over graphs. In addition, Dai & Wang (2021); Bose & Hamilton (2019); Fisher et al. (2020) propose to use adversarial regularizors to reduce bias in GNNs. Another approach is to utilize a Bayesian approach where the sensitive information is modeled in the prior distribution to enhance fairness over graphs (Buyl & De Bie, 2020). Furthermore, Ma et al. (2021) performs a PAC-Bayesian analysis and links the notion of subgroup generalization to accuracy disparity, and Zeng et al. (2021) proposes several strategies including GNN-based ones to reduce bias for the representations of heterogeneous information networks. Specifically for fairness-aware link prediction, while Buyl & De Bie (2021) introduces a regularizer, Li et al. (2020); Laclau et al. (2021) propose strategies that alter the adjacency matrix. With a specific consideration of individual fairness over graphs, Dong et al. (2021b) proposes a ranking-based framework. Another research direction in fairness-aware graph-based learning is to modify the graph structure to combat bias resulted from the graph connectivity (Agarwal et al., 2021; Spinelli et al., 2021; Kose & Shen, 2022; Köse & Shen, 2021). Differing from all previous works, the proposed framework herein proposes a unified framework that can mitigate bias in GNN-based learning together with an enhanced convergence speed.

**Normalization:** Batch Normalization (BatchNorm) (Ioffe & Szegedy, 2015) is the pioneering study that proposes to shift and scale the hidden representations in a batch to accelerate the convergence of training for DNNs. Following that, several normalization strategies are presented so far for different domains (Huang et al., 2020; Ioffe & Szegedy, 2015; Ulyanov et al., 2016; Ba et al., 2016; Salimans & Kingma, 2016; Xiong et al., 2020; Miyato et al., 2018; Wu & He, 2018; Santurkar et al., 2018; Xu et al., 2019a; Dong et al., 2021a; Huang et al., 2022). Specifically, LayerNorm is presented for natural language processing (Ba et al., 2016), and InstanceNorm (Ulyanov et al., 2016) seeks to improve the optimization for style transfer tasks, while Yi et al. (2018); Sun et al. (2020) target at permutation-equivalent data processing. Moreover, Chang et al. (2019) presents a domain-specific normalization technique for unsupervised domain adaptation, and Liu et al. (2020); Yu et al. (2020) provide normalization-based solutions for adversarial robustness and image inpainting, respectively.

For GNNs, Xu et al. (2018a) adapts BatchNorm within the framework of graph isomorphism networks, while a prior version of Dwivedi et al. (2020) normalizes node features based on the graph size. By taking the graph structure into account, Chen et al. (2022) proposes two novel normalization techniques, as well as an attention mechanism that learns a weighted combination of multiple graph-aware normalization strategies. A size-agnostic normalization for graphs, GraphNorm is proposed in (Cai et al., 2021), which improves InstanceNorm for graphs with a learnable shift to prevent degradation in expressiveness. Normalization frameworks are also utilized to combat the over-smoothing issue over GNNs, where Zhou et al. (2020) applies the normalization independently over clusters that are defined with respect to the class labels, and Zhao & Akoglu (2020) normalizes the total pairwise feature distances. It is important to note that the cluster-wise normalization introduced in Zhou et al. (2020) aims to improve the discrimination of the representations from different classes. However, the cluster definition therein differ significantly from the present work. Furthermore, none of the aforementioned normalization schemes consider fairness.

## 3 Preliminaries

This study develops a unified training scheme for GNNs that can improve fairness while at the same time enhance the convergence speed, given an input graph $\mathcal{G} := (\mathcal{V}, \mathcal{E})$, where $\mathcal{V} := \{v_1, v_2, \cdots, v_N\}$ denotes the node set, and $\mathcal{E} \subseteq \mathcal{V} \times \mathcal{V}$ is the edge set. Matrices $\mathbf{X} \in \mathbf{R}^{F \times N}$ and $\mathbf{A} \in \{0,1\}^{N \times N}$ are the feature and adjacency matrices, respectively, where $\mathbf{A}_{ij} = 1$ if and only if $(v_i, v_j) \in \mathcal{E}$ and $F$ is the dimension of features. Degree matrix $\mathbf{D} \in \mathbf{R}^{N \times N}$ is defined to be a diagonal matrix with the $n$th diagonal entry denoting the degree of node $v_n$. In this study, the sensitive attributes of the nodes are denoted by $\mathbf{s} \in \{0,1\}^N$, where the existence of a single, binary sensitive attribute is considered. Furthermore, $\mathcal{S}^0$ and $\mathcal{S}^1$ denote the set of nodes whose sensitive attributes are 0 and 1, respectively. Node representations at $k$th layer are represented by $\mathbf{H}^{(k)} \in \mathbf{R}^{F \times N}$, where $\mathbf{h}_j$ denotes the representation of node $v_j$ and $h_{i,j}$ is the $i$th feature of $\mathbf{h}_j$. Vectors $\mathbf{x}_j \in \mathbf{R}^F$ and $s_j \in \{0,1\}$ will be used to denote the feature vector and the sensitive attribute of node $v_j$. Throughout the paper, $\max(\cdot, \ldots, \cdot)$ outputs the element-wise maximum vector of its argument vectors, and $\mathrm{mean}(\cdot, \ldots, \cdot)$ denotes the sample mean operator.

GNNs learn node embeddings by repeatedly aggregating information from neighboring nodes. GNNs with different aggregation schemes have been developed, see in (Kipf & Welling, 2017; Veličković et al., 2018; Xu et al., 2018a). A general formulation of GNNs in the matrix form can be written as:

$$\mathbf{H}^{(k)} = \mathrm{Act}\left(\mathbf{W}^{(k)}\mathbf{H}^{(k-1)}\mathbf{Q}\right)$$

where $\mathbf{W}^{(k)}$ represents the weight matrix of GNN at $k$th layer and Act denotes activation function. In this formulation, $\mathbf{Q}$ matrix specifies the information aggregation process from neighbors, which changes in different GNN frameworks. For example, $\mathbf{Q} = \hat{\mathbf{D}}^{-\frac{1}{2}}\hat{\mathbf{A}}\hat{\mathbf{D}}^{-\frac{1}{2}}$ for Graph Convolutional Networks (GCN) (Kipf & Welling, 2017), where $\hat{\mathbf{A}} = \mathbf{A} + \mathbf{I}_N$ with $\mathbf{I}_N \in \{0,1\}^{N \times N}$ denoting the identity matrix, and $\hat{\mathbf{D}}$ is the degree matrix corresponding to $\hat{\mathbf{A}}$. Finally, the representations created after one aggregation process are denoted by $\mathbf{Z}^{(k)} = \mathbf{H}^{(k-1)}\mathbf{Q}$. Note that the superscript $(k)$ for layer number is dropped in the remaining of the paper, as the proposed framework is applicable to every layer in the same way.

### 3.1 Normalization for GNNs

While different normalization schemes have been proposed for different domains, there is not a universal normalization strategy that suits every domain (Cai et al., 2021). For GNNs, Cai et al. (2021) demonstrates that mean normalization can degrade the expressiveness of the neural networks, as mean statistics incorporate graph structural information. Motivated by this, Cai et al. (2021) proposes GraphNorm, which employs a learnable shift to preserve the mean statistics to a certain extent. The study reports that GraphNorm consistently achieves superior convergence speed and training stability on graph classification for GNNs over other normalization strategies. For the input matrix $\mathbf{A} \in \mathbf{R}^{F \times N}$, GraphNorm can be mathematically described as:

$$\text{GraphNorm}\left(a_{i,j}\right) = \gamma_i \cdot \frac{A_{i,j} - \alpha_i \cdot m_i}{\hat{\sigma}_i} + \beta_i, \forall i = 1, \dots F \tag{1}$$

where $m_i = \frac{\sum_{j=1}^N a_{i,j}}{N}, \hat{\sigma}_i^2 = \frac{\sum_{j=1}^N (a_{i,j} - \alpha_i \cdot m_i)^2}{N}$, and $\alpha_i, \gamma_i, \beta_i$ are the learnable parameters.

### 3.2 Bias in GNNs

ML models can lead to discriminatory results towards certain under-represented groups, as they propagate the bias within the training data (Dwork et al., 2012; Beutel et al., 2017). It has been demonstrated that the utilization of graph structure in GNNs amplifies the already existing bias (Dai & Wang, 2021). Thus, understanding the sources of bias in graph structure is crucial to develop a remedy for it. Motivated by this, Li et al. (2020) and Kose & Shen (2022) investigate the sources of bias in GNN-based learning. In Li et al. (2020), the representation discrepancy between different sensitive groups is examined, whereas in Kose & Shen (2022), the bias analysis is based on the correlation between the aggregated representations $\mathbf{Z}$ and sensitive attributes $\mathbf{s}$. Though through different approaches, both analyses in Li et al. (2020, Theorem 4.1) and Kose & Shen (2022, Theorem 3.1) demonstrate the parallelism between the terms $\|\boldsymbol{\mu}^{(0)} - \boldsymbol{\mu}^{(1)}\|, \|\boldsymbol{\Delta}\|$ and bias in GNN-based learning. Here, $\boldsymbol{\mu}^{(0)}$ and $\boldsymbol{\mu}^{(1)}$ are the sample means of node representations respectively across each sensitive group, where $\boldsymbol{\mu}^{(n)} = \text{mean}(\mathbf{h}_j \mid v_j \in \mathcal{S}^n)$, and $\boldsymbol{\Delta}$ stands for the maximal deviations of hidden representations, that is $\Delta_i^{(n)} = \max_j |h_{i,j}^{(n)} - \mu_i^{(n)}|, \forall i = 1, \cdots, F$ and $\boldsymbol{\Delta} = \max(\boldsymbol{\Delta}^{(0)}, \boldsymbol{\Delta}^{(1)})$. The superscript $(n)$ in $\mathbf{h}_j^{(n)}$ is utilized to specify the sensitive group index. Specifically, the hidden representation $\mathbf{h}_j^{(n)}$ corresponds to a node $v_j \in \mathcal{S}^n$.

As the analyses in Li et al. (2020); Kose & Shen (2022) suggest that the distributions of hidden representations corresponding to different sensitive groups influence the resulted bias by GNNs, a tool that can shift these group-wise distributions can effectively decrease bias-related terms, and hence the overall bias.

## 4 FairNorm: A Fair and Fast Training Framework for GNNs

This section presents the proposed unified framework that achieves fairness improvement together with faster convergence speed for GNN-based learning. Herein, we first present a group-wise normalization framework (M-Norm), upon which we develop FairNorm by incorporating two novel fairness regularizers on M-norm for fairness-enhancement.

### 4.1 Group-wise Normalization

It has been demonstrated in Li et al. (2020); Kose & Shen (2022) that decreasing $\|\boldsymbol{\mu}^{(0)} - \boldsymbol{\mu}^{(1)}\|$ and $\boldsymbol{\Delta}$ can effectively reduce bias in GNN-based learning. Note that both terms are affected by the distributions of representations from different sensitive groups. On the other hand, the mean and standard deviation of the hidden representations, and in turn their distributions, are affected by the learnable parameters of a normalization layer. Thus, employing such a layer can enable manipulating said distributions, which can be used to improve fairness. Inspired by this, the proposed framework, FairNorm, first applies normalizations to different sensitive groups individually, which results in individual learnable parameters affecting $\boldsymbol{\mu}^{(0)}$, and $\boldsymbol{\mu}^{(1)}$, as well as their difference. For any input matrix $\mathbf{A} \in \mathbb{R}^{F \times N}$, given that the columns of $\mathbf{A}$ can be divided into two sensitive groups $\mathcal{S}^0$ and $\mathcal{S}^1$, the corresponding **m**ultiple group-wise **norm**alization operations can

be mathematically described as:

$$\text{M-Norm}\left(a_{i,j}^{(n)}\right) = \gamma_i^{(n)} \cdot \frac{A_{i,j}^{(n)} - \alpha_i^{(n)} \cdot m_i^{(n)}}{\sigma_i^{(n)}} + \beta_i^{(n)}, \tag{2}$$

where $m_i^{(n)} = \frac{\sum_{j=1}^{|\mathcal{S}^n|} a_{i,j}^{(n)}}{|\mathcal{S}^n|}, (\sigma_i^{(n)})^2 = \frac{\sum_{j=1}^{|\mathcal{S}^n|}\left(a_{i,j}^{(n)} - m_i^{(n)}\right)^2}{|\mathcal{S}^n|}$, and $\alpha_i^{(n)}, \gamma_i^{(n)}, \beta_i^{(n)}$ are learnable parameters, $\forall i = 1, \ldots, F$ and $n = 0, 1$. The superscript $(n)$ in $\mathbf{A}_j^{(n)}$ specifies that the representation corresponds to a node from the sensitive group $\mathcal{S}^n$. Considering that mean normalization can degrade the expressiveness of GNNs (Cai et al., 2021), the proposed framework employs the learnable parameter $\boldsymbol{\alpha}$ that manages the amount of mean normalization.

It is demonstrated in Cai et al. (2021) that applying a shift operation over the whole graph can speed up the convergence for graph classification. However, as the proposed framework herein applies multiple shift operations individually over subgraphs corresponding to different sensitive groups and considers the node classification task, the effect of the proposed strategy on the convergence speed becomes unclear. Hence, the analysis in Cai et al. (2021) cannot be directly applied to this case. Motivated by this, this study analytically examines the influence of group-wise shifts on the convergence speed.

Shift operations over different sensitive groups can be applied in matrix forms via the matrices $\mathbf{N}^{(0)}$ and $\mathbf{N}^{(1)}$, where $\mathbf{N}^{(n)} = \mathbf{I}_N - \frac{1}{|\mathcal{S}^n|}\mathbf{e}^{(n)}(\mathbf{e}^{(n)})^\top$ for $n = 0, 1$. In this formulation, $\mathbf{e}^{(n)} \in \mathbb{R}^N$ is created such that $e_j^{(n)} = 1$ if $v_j \in \mathcal{S}^n$, and $e_j^{(n)} = 0$ otherwise. Therefore, for any vector $\mathbf{c} \in \mathbb{R}^N$, $\mathbf{c}^\top\mathbf{N}^{(n)} = \mathbf{c}^\top - (\frac{1}{|\mathcal{S}^n|}\sum_{j:v_j\in\mathcal{S}^n} c_j)(\mathbf{e}^{(n)})^\top$. Hence, the group-wise shift operations applied to hidden representations can be written as:

$$\text{MShift}(\mathbf{W}^{(k)}\mathbf{H}^{(k-1)}\mathbf{Q}) = \mathbf{W}^{(k)}\mathbf{H}^{(k-1)}\mathbf{Q}\mathbf{N}^{(0)}\mathbf{N}^{(1)}. \tag{3}$$

The following lemma demonstrates that $\mathbf{N}^{(0)}\mathbf{N}^{(1)}$ acts as a preconditioner of $\mathbf{Q}$, whose proof is presented in Appendix A.

**Lemma 1** *Let $\lambda_i$ and $\sigma_i$ denote the ith singular values of $\mathbf{Q}$ and $\mathbf{Q}\mathbf{N}^{(0)}\mathbf{N}^{(1)}$, respectively. It can be shown that $\mathbf{Q}\mathbf{N}^{(0)}\mathbf{N}^{(1)}$ has at least two singular values being equal to zero, i.e., $\gamma_N = \gamma_{N-1} = 0$. Without loss of generalization, assume $0 \leqslant \lambda_1 \leqslant \cdots \leqslant \lambda_N$ and $0 \leqslant \gamma_1 \leqslant \gamma_2 \leqslant \cdots \leqslant \gamma_{N-2}$. Then, the following holds:*

$$\lambda_1 \leqslant \gamma_1$$
$$\lambda_2 \leqslant \gamma_2$$
$$\vdots$$
$$\lambda_{N-2} \leqslant \gamma_{N-2} \leqslant \lambda_N, \tag{4}$$

*where the equalities hold, i.e., $\lambda_i = \gamma_i$ or $\lambda_N = \gamma_{N-2}$, only if $\mathbf{Q}$ has a right singular vector $\boldsymbol{\alpha}$ for which $(\mathbf{e}^{(0)})^\top\boldsymbol{\alpha} = 0$ and $(\mathbf{e}^{(1)})^\top\boldsymbol{\alpha} = 0$ are satisfied.*

In other domains, such as DNNs or iterative algorithms, a similar preconditioning is considered to help the training (Kingma & Ba, 2014; Axelsson, 1985). Such a preconditioning of the aggregation matrix $\mathbf{Q}$ is also demonstrated to accelerate the optimization of GNNs (Cai et al., 2021). In order to theoretically investigate such an effect in our setting, we considered a basic linear GNN model for node classification that is optimized via gradient descent, and presented its convergence analysis in Theorem 1. Appendix B presents all assumptions and considered learning settings employed in Theorem 1 in detail, as well as its proof.

**Theorem 1** *Let $\mathbf{w}_t^{Vanilla}$ and $\mathbf{w}_t^{MShift}$ denote the parameters of a linear GNN model at time t for the cases where no shift is applied and shift operations $\mathbf{N}^{(0)}\mathbf{N}^{(1)}$ are applied, respectively. It holds with high probability that*

$$\left\|\mathbf{w}_t^{MShift} - \mathbf{w}_*^{MShift}\right\|_2 = O\left(\rho_1^t\right) \quad \text{and} \quad \left\|\mathbf{w}_t^{Vanilla} - \mathbf{w}_*^{Vanilla}\right\|_2 = O\left(\rho_2^t\right), \quad \text{where } \rho_1 < \rho_2. \tag{5}$$

*Here, $\mathbf{w}_*^{Vanilla}$ and $\mathbf{w}_*^{MShift}$ denote the optimal values for $\mathbf{w}_t^{Vanilla}$ and $\mathbf{w}_t^{MShift}$, respectively. Thus, it concludes that the shift operations applied through $\mathbf{N}^{(0)}\mathbf{N}^{(1)}$ lead to faster convergence with high probability compared to the scheme where no shift is applied.*

Theorem 1 demonstrates that the individual shift operations applied over different sensitive groups indeed improve the convergence rate compared to the naive baseline. While the result of Theorem 1 seems to be similar to the result of Cai et al. (2021, Proposition 3.1), the analysis in Cai et al. (2021) cannot be easily extended to our proof due to the employment of group-wise shifts in this work and the fact that we consider node classification instead of graph classification.

## 4.2 Fairness-aware Regularizers

As a complementary step to M-Norm, FairNorm introduces two novel fairness-aware regularizers to mitigate bias in GNN-based learning. Consider the conventional case where the normalization is applied after linear transformations (Ioffe & Szegedy, 2015; Xiong et al., 2020; Cai et al., 2021). For this case, the hidden representations can be expressed in matrix form as:

$$\mathbf{H}^{(n)} = \text{Act}\left(\text{M-Norm}^{(n)}\left((\mathbf{WHQ})^{(n)}\right)\right), \text{ for } n = 0, 1. \tag{6}$$

In equation 6, $(\mathbf{WHQ})^{(n)}$ denotes the submatrix consisting the columns of $(\mathbf{WHQ})$ whose corresponding nodes are in $\mathcal{S}^n$. Furthermore, as the proposed strategy applies normalizations individually over different sensitive groups, these group-wise normalization layers are differentiated by the superscript $n = 0, 1$. Let $\bar{\boldsymbol{\mu}}^{(n)} \in \mathbb{R}^F$ denote the sample mean of representations after normalization for the sensitive group $\mathcal{S}^n, n = 0, 1$. In the proposed framework, recalling from Subsection 4.1, individual normalization layers are employed to create individual learnable parameters for the distributions of different sensitive groups, so that the bias-related terms derived in Li et al. (2020); Kose & Shen (2022) can be reduced. However, in order to manipulate $\bar{\boldsymbol{\mu}}^{(0)}$ and $\bar{\boldsymbol{\mu}}^{(1)}$ for possible bias reduction, the relationship between $\|\boldsymbol{\mu}^{(0)} - \boldsymbol{\mu}^{(1)}\|$ and $\bar{\boldsymbol{\mu}}^{(n)}$'s should be investigated. To this end, we present the following theorem, the proof of which can be found in Appendix C.

**Theorem 2** *Let* $\text{Act}(.)$ *be Lipschitz continuous with Lipschitz constant* $L$, *and let* $\bar{\mathbf{H}}^{(n)}$ *denote the normalized representations in group* $S^n$. *Then,* $\|\boldsymbol{\mu}^{(0)} - \boldsymbol{\mu}^{(1)}\|$ *is bounded above by*

$$\|\boldsymbol{\mu}^{(0)} - \boldsymbol{\mu}^{(1)}\|_p \leqslant L\left(\|\bar{\boldsymbol{\mu}}^{(0)} - \bar{\boldsymbol{\mu}}^{(1)}\|_p + \|\bar{\boldsymbol{\Delta}}^{(0)}\|_p + \|\bar{\boldsymbol{\Delta}}^{(1)}\|_p\right), \quad \forall p \geqslant 1. \tag{7}$$

*Here,* $\bar{\boldsymbol{\Delta}}^{(n)}$ *is the maximal deviation of* $\bar{\mathbf{H}}^{(n)}$ *from* $\bar{\boldsymbol{\mu}}^{(n)}$ *(i.e., we have* $\bar{\Delta}_i^{(n)} = \max_j |\bar{h}_{i,j}^{(n)} - \bar{\mu}_i^{(n)}|, \forall i = 1, \cdots, F$*).*

Theorem 2 demonstrates that decreasing $\|\bar{\boldsymbol{\mu}}^{(0)} - \bar{\boldsymbol{\mu}}^{(1)}\|$ results in a decreased upper bound for $\|\boldsymbol{\mu}^{(0)} - \boldsymbol{\mu}^{(1)}\|$, which can possibly reduce the actual value of $\|\boldsymbol{\mu}^{(0)} - \boldsymbol{\mu}^{(1)}\|$. Based on this result, as a second step after applying group-wise normalization operators, FairNorm proposes the use of a regularizer term $\mathcal{L}_\mu = \|\bar{\boldsymbol{\mu}}^{(0)} - \bar{\boldsymbol{\mu}}^{(1)}\|_2^2$ to decrease bias for GNN-based learning. We note that many commonly used activation functions such as ReLU, sigmoid, tanh, *etc.* have a Lipschitz constant equal to $L = 1$.

Furthermore, Theorem 2 shows that the upper bound for $\|\boldsymbol{\mu}^{(0)} - \boldsymbol{\mu}^{(1)}\|$ can also be decreased by reducing the norms of maximal deviations $\bar{\boldsymbol{\Delta}}^{(0)}$ and $\bar{\boldsymbol{\Delta}}^{(1)}$. Inspired by this finding, $\mathcal{L}_\Delta = \|\bar{\boldsymbol{\Delta}}^{(0)}\|_2^2 + \|\bar{\boldsymbol{\Delta}}^{(1)}\|_2^2$ is also introduced as a regularizer to reduce the norms of maximal deviations of the normalized representations. Hence, the overall learning objective for the considered node classification task can be written as:

$$\min_{\theta_{GNN}} \mathcal{L}_c + \kappa \mathcal{L}_\mu + \tau \mathcal{L}_\Delta \tag{8}$$

where $\mathcal{L}_c$ is the classification loss, $\mathcal{L}_\mu = \|\bar{\boldsymbol{\mu}}^{(0)} - \bar{\boldsymbol{\mu}}^{(1)}\|_2^2 = \|\left(\boldsymbol{\gamma}^{(0)}\frac{\mathbf{m}^{(0)}(\mathbf{1}-\boldsymbol{\alpha}^{(0)})}{\boldsymbol{\sigma}^{(0)}} + \boldsymbol{\beta}^{(0)}\right) - \left(\boldsymbol{\gamma}^{(1)}\frac{\mathbf{m}^{(1)}(\mathbf{1}-\boldsymbol{\alpha}^{(1)})}{\boldsymbol{\sigma}^{(1)}} + \boldsymbol{\beta}^{(1)}\right)\|_2^2$ for M-Norm in equation 2. $\mathcal{L}_\Delta = \|\bar{\boldsymbol{\Delta}}^{(0)}\|_2^2 + \|\bar{\boldsymbol{\Delta}}^{(1)}\|_2^2$, and $\bar{\Delta}_i^{(n)} = \frac{\gamma_i^{(n)}}{\sigma_i^{(n)}}\max_j|r_{i,j}^{(n)} - m_i^{(n)}|$ for M-Norm defined in equation 2 $\forall i = 1, \cdots, F$, where $\mathbf{R}^{(n)} = (\mathbf{WHQ})^{(n)}$ denotes the representations input to the normalization layer and $r_{i,j}^{(n)}$ is the element in $i$th row and $j$th column of $\mathbf{R}^{(n)}$ matrix. GNN parameters are denoted by $\theta_{GNN}$, and $\kappa$ and $\tau$ are hyperparameters specifying

the focus on the fairness regularizers.

**Remark 1 (Order of normalization and activation).** Although the proposed fairness regularizers $\mathcal{L}_\mu, \mathcal{L}_\Delta$ are designed for the conventional case where the normalization is used before nonlinear activation, it can be demonstrated that they can also reduce bias when the normalization is applied after activation, where

$$\mathbf{H}^{(n)} = \text{M-Norm}^{(n)} \left( \text{Act} \left( (\mathbf{WHQ})^{(n)} \right) \right), \text{ for } n = 0, 1. \tag{9}$$

In this case, it holds that $\bar{\boldsymbol{\mu}}^{(n)} = \boldsymbol{\mu}^{(n)}$ and $\|\bar{\boldsymbol{\Delta}}^{(n)}\| = \|\boldsymbol{\Delta}^{(n)}\|$ for $n = 0, 1$. Therefore, the employment of the proposed fairness regularizers can naturally be extended to the case where the normalization is utilized after activation, as the analyses in Li et al. (2020); Kose & Shen (2022) demonstrate that reducing $\|\boldsymbol{\mu}^{(0)} - \boldsymbol{\mu}^{(1)}\|$ and $\|\boldsymbol{\Delta}\|$ can help mitigate bias in GNN-based learning.

**Remark 2 (Applicability to other normalization methods).** Note that the proposed FairNorm framework can be readily utilized together with other normalization techniques where the distribution of the normalized representations depends on learnable parameters, e.g., BatchNorm (Ioffe & Szegedy, 2015), as long as group-wise normalization is applied over each sensitive group.

Table 1: Dataset statistics

| Dataset | $|\mathcal{S}_0|$ | $|\mathcal{S}_1|$ | Inter-edges | Intra-edges | $F$ |
|---|---|---|---|---|---|
| Pokec-z | 4851 | 2808 | 1140 | 28336 | 59 |
| Pokec-n | 4040 | 2145 | 943 | 20901 | 59 |
| Recidivism | 9317 | 9559 | 298098 | 325642 | 17 |

## 5 Experiments

In this section, experimental results obtained on real-world datasets for a supervised node classification task are presented. The performance of the proposed framework, FairNorm, is compared with baseline schemes in terms of node classification accuracy and fairness metrics. Furthermore, the influence of the proposed fairness-aware normalization strategy on convergence speed is examined.

### 5.1 Datasets and Settings

**Datasets.** In the experiments, three real-world networks are used: Pokec-z, Pokec-n (Dai & Wang, 2021), and the Recidivism graph (Jordan & Freiburger, 2015). Pokec-z and Pokec-n are created by sampling the anonymized, 2012 version of Pokec (Takac & Zabovsky, 2012), which is a social network used in Slovakia (Dai & Wang, 2021). In Pokec networks, the region information is utilized as the sensitive attribute, where the nodes are the users living in two major regions. Labels to be used in node classification are assigned to be the binarized working field of the users. The information of defendants (corresponding to nodes) who got released on bail at the U.S. state courts during 1990-2009 (Jordan & Freiburger, 2015) is utilized to build the Recidivism graph, where the edges are formed based on the similarity of past criminal records and demographics. Race is used as the sensitive attribute for this graph, and the node classification task classifies defendants into bail (i.e., the defendant is not likely to commit a violent crime if released) or no bail (i.e., the defendant is likely to commit a violent crime if released) (Agarwal et al., 2021). Table 1 presents further statistical information on the utilized datasets. In the table, $|\mathcal{S}_0|$ and $|\mathcal{S}_1|$ are the cardinalities of the sets of nodes with sensitive attributes 0 and 1, respectively. 'Inter-edges' and 'Intra-edges' correspond to the number of edges linking nodes from different sensitive groups and the same sensitive group, respectively. $F$ in Table 1 denotes the dimension of nodal features.

**Evaluation Metrics.** Accuracy is used as the utility measure for node classification. Two quantitative measures of group fairness metrics are reported in terms of *statistical parity*: $\Delta_{SP} = |P(\hat{y} = 1 \mid s = 0) - P(\hat{y} = 1 \mid s = 1)|$ and *equal opportunity*: $\Delta_{EO} = |P(\hat{y} = 1 \mid y = 1, s = 0) - P(\hat{y} = 1 \mid y = 1, s = 1)|$, where $y$ is the ground truth label, and $\hat{y}$ denotes the predicted label. Lower values for $\Delta_{SP}$ and $\Delta_{EO}$ signify better fairness performance (Dai & Wang, 2021).

**Implementation details.** To comparatively evaluate our proposed framework, node classification is utilized in a supervised setting. A two-layer GCN (Kipf & Welling, 2017) followed by a linear layer is employed for the classification task, which is identical to the experimental setting used in Dai & Wang (2021). A normalization layer follows after every GNN layer, where the normalization is applied after linear transformations and before the non-linear activation, as suggested in Ioffe & Szegedy (2015); Cai et al. (2021). For the hyperparameter selection of the GCN model, see Appendix D. This experimental framework is kept the same for all baselines. Furthermore, training of the model is executed over 50% of the nodes, while the remaining nodes are equally divided to be used as the validation and test sets. For each experiment, results for five random data splits are obtained, and the average of them together with standard deviations are presented. The hyperparameters of the proposed fairness-aware framework and all other baselines are tuned via a grid search on cross-validation sets, see again Appendix D for the utilized hyperparameter values.

**Baselines.** This work aims to mitigate bias via employing fairness-aware regularizers, as well as to provide a faster convergence through its utilized normalization layers. We note that similar to the proposed regularizers, any other fairness-aware regularizer can be employed together with a normalization layer, for these same purposes. In order to demonstrate the performance improvement of the proposed regularizers over said alternatives, we compare the proposed framework with other fairness-aware regularizers. To this end, the performance of 4 different baselines is presented. For improving fairness in a supervised setting, FairGNN (Dai & Wang, 2021) employs adversarial debiasing and a covariance-based regularizer (the absolute covariance between the sensitive attribute and estimated label $\hat{\mathbf{y}}$). The results for these regularizers are obtained both individually and together, where the framework that utilizes both regularizers is called FairGNN (Dai & Wang, 2021). Furthermore, hyperbolic tangent relaxation of the difference of demographic parity ($HTR_{DDP}$) that is proposed in Padh et al. (2021) is utilized as another baseline. Note that, as DDP is not differentiable, its relaxations are used as fairness-aware regularizers for a gradient-based optimization. It is worth emphasizing that the fairness regularizers proposed in this study are also applicable to an unsupervised setting, while the covariance-based (also FairGNN) and $HTR_{DDP}$ regularizers can only be used in a supervised framework.

## 5.2 Experimental Results

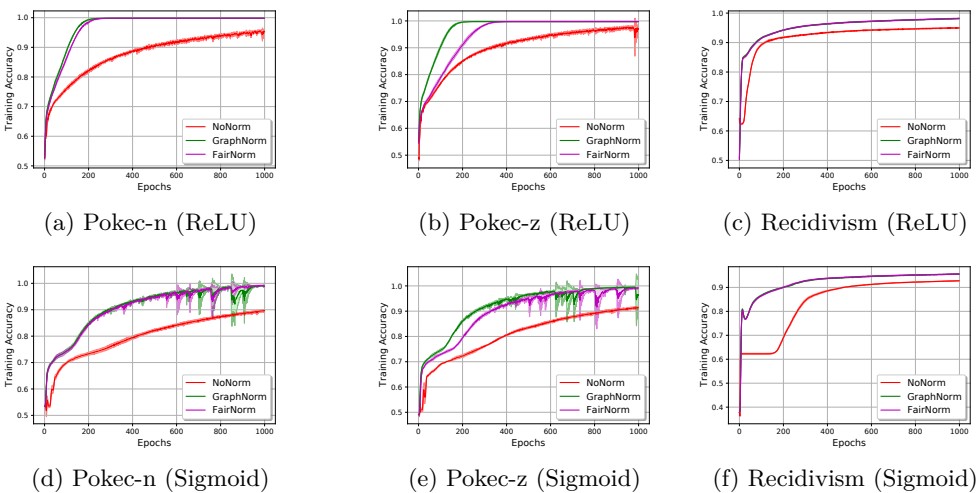

Figure 1: Convergence for different graph data sets when the normalization is not applied (Nonorm) and applied with/without fairness consideration (FairNorm/GraphNorm).

The results of node classification are presented in Table 2 in terms of fairness and utility metrics for both the proposed framework and baselines. The results are obtained for two commonly utilized activation functions: ReLU and sigmoid, in order to demonstrate the efficacy of the proposed framework over different activations. In Table 2, "NoNorm" denotes the scheme where no normalization layer is employed. "M-Norm" stands for the proposed framework where only individual normalizations are applied to the nodes belonging to different

Table 2: Comparative Results with Baselines for Different Activation Function Selections

| | ReLU | | | Sigmoid | | |
|---|---|---|---|---|---|---|
| **Pokec-z** | Acc (%) | $\Delta_{SP}$ (%) | $\Delta_{EO}$ (%) | Acc (%) | $\Delta_{SP}$ (%) | $\Delta_{EO}$(%) |
| NoNorm | $70.24 \pm 1.0$ | $6.77 \pm 1.8$ | $6.18 \pm 2.5$ | $\mathbf{70.25} \pm 0.8$ | $7.40 \pm 1.8$ | $6.04 \pm 3.1$ |
| M-Norm | $\mathbf{70.71} \pm 0.8$ | $5.57 \pm 1.3$ | $5.00 \pm 2.0$ | $69.84 \pm 0.7$ | $6.21 \pm 1.4$ | $4.44 \pm 2.1$ |
| Covariance | $70.66 \pm 0.8$ | $5.31 \pm 1.4$ | $4.56 \pm 1.9$ | $69.77 \pm 0.6$ | $5.63 \pm 1.9$ | $4.04 \pm 2.1$ |
| Adversarial | $70.35 \pm 0.9$ | $2.41 \pm 1.0$ | $2.16 \pm 0.6$ | $70.01 \pm 0.9$ | $3.08 \pm 2.8$ | $3.00 \pm 2.4$ |
| FairGNN | $70.34 \pm 1.1$ | $2.78 \pm 1.5$ | $2.73 \pm 1.0$ | $69.93 \pm 0.7$ | $4.55 \pm 2.2$ | $4.71 \pm 2.7$ |
| $HTR_{DDP}$ | $70.38 \pm 0.9$ | $2.12 \pm 2.0$ | $3.38 \pm 1.5$ | $69.74 \pm 0.7$ | $1.85 \pm 1.1$ | $2.27 \pm 1.8$ |
| FairNorm | $70.67 \pm 1.0$ | $\mathbf{1.35} \pm 1.2$ | $\mathbf{1.90} \pm 1.8$ | $69.73 \pm 0.9$ | $\mathbf{1.71} \pm 0.3$ | $\mathbf{1.48} \pm 1.1$ |
| **Pokec-n** | Acc (%) | $\Delta_{SP}$ (%) | $\Delta_{EO}$ (%) | Acc (%) | $\Delta_{SP}$ (%) | $\Delta_{EO}$(%) |
| NoNorm | $69.29 \pm 0.8$ | $1.66 \pm 1.6$ | $2.19 \pm 1.8$ | $68.73 \pm 0.6$ | $2.68 \pm 2.2$ | $2.27 \pm 2.4$ |
| M-Norm | $69.25 \pm 0.5$ | $2.48 \pm 1.2$ | $2.91 \pm 1.7$ | $68.59 \pm 0.9$ | $1.78 \pm 2.1$ | $2.88 \pm 1.8$ |
| Covariance | $69.47 \pm 0.6$ | $2.06 \pm 1.3$ | $2.42 \pm 1.5$ | $68.40 \pm 1.1$ | $1.70 \pm 2.2$ | $2.26 \pm 1.8$ |
| Adversarial | $69.30 \pm 0.4$ | $2.09 \pm 1.9$ | $2.21 \pm 1.9$ | $68.47 \pm 0.7$ | $1.56 \pm 1.8$ | $2.25 \pm 1.5$ |
| FairGNN | $69.21 \pm 0.4$ | $2.03 \pm 1.9$ | $2.29 \pm 2.1$ | $68.42 \pm 0.7$ | $1.61 \pm 1.7$ | $\mathbf{1.71} \pm 2.0$ |
| $HTR_{DDP}$ | $\mathbf{69.51} \pm 0.5$ | $1.85 \pm 1.4$ | $2.03 \pm 1.5$ | $68.37 \pm 0.9$ | $1.64 \pm 1.6$ | $2.53 \pm 1.7$ |
| FairNorm | $69.38 \pm 0.7$ | $\mathbf{1.26} \pm 1.2$ | $\mathbf{1.22} \pm 1.3$ | $\mathbf{68.88} \pm 1.1$ | $\mathbf{1.44} \pm 1.2$ | $1.74 \pm 1.7$ |
| **Recidivism** | Acc (%) | $\Delta_{SP}$ (%) | $\Delta_{EO}$ (%) | Acc (%) | $\Delta_{SP}$ (%) | $\Delta_{EO}$(%) |
| NoNorm | $94.32 \pm 0.2$ | $8.89 \pm 0.7$ | $1.17 \pm 0.9$ | $92.69 \pm 0.2$ | $8.29 \pm 0.7$ | $1.31 \pm 0.6$ |
| M-Norm | $95.00 \pm 0.3$ | $8.87 \pm 1.2$ | $1.71 \pm 0.7$ | $\mathbf{94.45} \pm 0.3$ | $8.94 \pm 1.2$ | $2.06 \pm 1.0$ |
| Covariance | $95.07 \pm 0.2$ | $8.82 \pm 1.1$ | $1.43 \pm 0.6$ | $92.87 \pm 1.0$ | $8.44 \pm 0.6$ | $1.64 \pm 1.0$ |
| Adversarial | $94.14 \pm 0.1$ | $8.58 \pm 1.0$ | $1.26 \pm 0.7$ | $93.82 \pm 0.2$ | $8.72 \pm 0.9$ | $1.59 \pm 1.1$ |
| FairGNN | $95.14 \pm 0.2$ | $8.73 \pm 1.0$ | $1.33 \pm 0.8$ | $94.11 \pm 0.2$ | $8.68 \pm 1.2$ | $1.51 \pm 0.5$ |
| $HTR_{DDP}$ | $\mathbf{95.16} \pm 0.2$ | $8.74 \pm 0.8$ | $1.05 \pm 0.4$ | $93.34 \pm 0.4$ | $8.20 \pm 0.9$ | $1.04 \pm 1.0$ |
| FairNorm | $95.11 \pm 0.2$ | $\mathbf{8.45} \pm 1.0$ | $\mathbf{0.90} \pm 0.5$ | $94.32 \pm 0.2$ | $\mathbf{7.28} \pm 1.1$ | $\mathbf{0.80} \pm 0.9$ |

sensitive groups, without using the proposed fairness regularizers. Furthermore, "Covariance" is for the covariance-based regularizer (Dai & Wang, 2021), "Adversarial" stands for the adversarial regularizer (Dai & Wang, 2021), and "$HTR_{DDP}$" denotes hyperbolic tangent relaxation of the difference of demographic parity (Padh et al., 2021). It should be noted that the results for baselines are obtained with the best performing normalization layer framework (individual normalizations over different sensitive groups vs. normalization over all nodes in the graph) in terms of fairness measures.

The results in Table 2 demonstrate that FairNorm achieves superior fairness performance, together with similar utility, compared to all baselines on all datasets for both of the utilized activations. As M-Norm is the first step of FairNorm without fairness regularizers, the significant improvements in all fairness measures compared to M-Norm signify the efficiency of the designed regularizers. Furthermore, on the Recidivism graph with sigmoid activation, while the improvement in fairness metrics is accompanied by a decrease in accuracy for the baselines, FairNorm achieves better fairness performance without a deterioration in utility. Overall, the results in Table 2 show the efficacy of the proposed fairness regularizers in reducing bias while providing similar utility on different real-world networks. Note that, in addition to their superior fairness performance, the proposed regularizers of FairNorm can be flexibly applied to both supervised and unsupervised settings, whereas some of the baselines ("Covariance", "FairGNN", "$HTR_{DDP}$") require predicted labels for their regularizer designs.

The proposed framework herein aims to mitigate bias by also providing a faster convergence speed. The results in Table 2 confirm that the proposed fairness regularizers within FairNorm do provide said bias reduction. In order to evaluate the convergence speed of FairNorm's group-wise normalizations, Figure 1 is presented. The baselines in Figure 1 consist of GraphNorm (Cai et al., 2021), and the framework where no normalization is applied. We note that in Figure 1, Fairnorm is employed with both its individual normalizations as well as its fairness regularizers.

The results on both Pokec datasets and the Recidivism network confirm that the employed normalization can indeed lead to a faster convergence in training compared to NoNorm. Figure 1 also demonstrates that compared to GraphNorm, the convergence improvement of FairNorm is slightly less on Pokec-z, whereas it provides approximately the same improvement on Pokec-n and Recidivism.

### 5.3 Sensitivity Analysis

Table 3: Sensitivity Analysis for Pokec Networks

| | Pokec-z | | | Pokec-n | | |
|---|---|---|---|---|---|---|
| **ReLU** | Acc (%) | $\Delta_{SP}$ (%) | $\Delta_{EO}$ (%) | Acc (%) | $\Delta_{SP}$ (%) | $\Delta_{EO}$(%) |
| M-Norm | **70.71** $\pm$ 0.8 | 5.57 $\pm$ 1.3 | 5.00 $\pm$ 2.0 | 69.25 $\pm$ 0.5 | 2.48 $\pm$ 1.2 | 2.91 $\pm$ 1.7 |
| $\tau = 10^{-8}, \kappa = 10$ | 70.26 $\pm$ 0.9 | 1.52 $\pm$ 1.0 | 2.19 $\pm$ 1.2 | 69.45 $\pm$ 0.6 | 2.23 $\pm$ 1.3 | 2.65 $\pm$ 1.6 |
| $\tau = 10^{-8}, \kappa = 100$ | 70.48 $\pm$ 0.9 | 1.40 $\pm$ 1.0 | 2.20 $\pm$ 1.2 | 69.53 $\pm$ 0.8 | 2.07 $\pm$ 1.0 | 1.98 $\pm$ 0.8 |
| $\tau = 10^{-8}, \kappa = 1000$ | 70.37 $\pm$ 1.1 | 3.20 $\pm$ 0.9 | 3.18 $\pm$ 2.0 | 69.38 $\pm$ 0.8 | **1.55** $\pm$ 1.4 | **1.48** $\pm$ 1.7 |
| $\kappa = 100, \tau = 10^{-7}$ | 70.67 $\pm$ 1.0 | **1.35** $\pm$ 1.2 | **1.90** $\pm$ 1.8 | **69.56** $\pm$ 0.8 | 2.10 $\pm$ 0.9 | 2.09 $\pm$ 0.5 |
| $\kappa = 100, \tau = 10^{-8}$ | 70.48 $\pm$ 0.9 | 1.40 $\pm$ 1.0 | 2.20 $\pm$ 1.7 | 69.53 $\pm$ 0.8 | 2.07 $\pm$ 1.0 | 1.98 $\pm$ 0.8 |
| $\kappa = 100, \tau = 10^{-9}$ | 70.55 $\pm$ 0.9 | 1.48 $\pm$ 1.2 | 1.99 $\pm$ 1.8 | 69.54 $\pm$ 0.7 | 2.15 $\pm$ 1.1 | 1.84 $\pm$ 0.9 |
| | Pokec-z | | | Pokec-n | | |
| **Sigmoid** | Acc (%) | $\Delta_{SP}$ (%) | $\Delta_{EO}$ (%) | Acc (%) | $\Delta_{SP}$ (%) | $\Delta_{EO}$(%) |
| M-Norm | 69.84 $\pm$ 0.7 | 6.21 $\pm$ 1.4 | 4.44 $\pm$ 2.1 | 68.59 $\pm$ 0.9 | 1.78 $\pm$ 2.1 | 2.88 $\pm$ 1.8 |
| $\tau = 10^{-8}, \kappa = 10$ | 69.71 $\pm$ 0.9 | 1.36 $\pm$ 0.4 | **1.72** $\pm$ 1.1 | **68.97** $\pm$ 0.8 | 2.23 $\pm$ 1.3 | 2.49 $\pm$ 1.8 |
| $\tau = 10^{-8}, \kappa = 100$ | 70.05 $\pm$ 1.2 | 1.19 $\pm$ 0.8 | 2.53 $\pm$ 1.0 | 68.88 $\pm$ 1.1 | 1.44 $\pm$ 1.2 | **1.74** $\pm$ 1.7 |
| $\tau = 10^{-8}, \kappa = 1000$ | **70.06** $\pm$ 1.1 | **1.17** $\pm$ 0.7 | 2.47 $\pm$ 1.3 | 68.37 $\pm$ 0.7 | 2.21 $\pm$ 1.1 | 3.47 $\pm$ 1.8 |
| $\kappa = 100, \tau = 10^{-7}$ | **70.06** $\pm$ 1.2 | 1.28 $\pm$ 0.8 | 2.39 $\pm$ 1.2 | 68.87 $\pm$ 1.1 | **1.45** $\pm$ 1.2 | 1.85 $\pm$ 1.6 |
| $\kappa = 100, \tau = 10^{-8}$ | **70.05** $\pm$ 1.2 | 1.19 $\pm$ 0.8 | 2.53 $\pm$ 1.0 | 68.88 $\pm$ 1.1 | **1.44** $\pm$ 1.2 | **1.74** $\pm$ 1.7 |
| $\kappa = 100, \tau = 10^{-9}$ | **70.06** $\pm$ 1.2 | 1.20 $\pm$ 0.8 | 2.53 $\pm$ 1.0 | 68.81 $\pm$ 1.1 | 1.70 $\pm$ 1.2 | 1.90 $\pm$ 1.7 |

Table 4: Sensitivity Analysis for Recidivism Network

| | ReLU | | | Sigmoid | | |
|---|---|---|---|---|---|---|
| **Recidivism** | Acc (%) | $\Delta_{SP}$ (%) | $\Delta_{EO}$ (%) | Acc (%) | $\Delta_{SP}$ (%) | $\Delta_{EO}$(%) |
| M-Norm | 95.00 $\pm$ 0.3 | 8.87 $\pm$ 1.2 | 1.71 $\pm$ 0.7 | **94.45** $\pm$ 0.3 | 8.94 $\pm$ 1.2 | 2.06 $\pm$ 1.0 |
| $\tau = 10^{-10}, \kappa = 1$ | 95.01 $\pm$ 0.2 | 8.76 $\pm$ 1.2 | 1.66 $\pm$ 0.6 | 94.38 $\pm$ 0.2 | **7.27** $\pm$ 1.0 | 0.95 $\pm$ 1.0 |
| $\tau = 10^{-10}, \kappa = 0.1$ | 95.02 $\pm$ 0.3 | 8.62 $\pm$ 1.1 | 1.43 $\pm$ 0.5 | 94.27 $\pm$ 0.3 | 7.29 $\pm$ 1.1 | 0.89 $\pm$ 1.0 |
| $\tau = 10^{-10}, \kappa = 0.01$ | **95.11** $\pm$ 0.2 | **8.45** $\pm$ 1.0 | **0.90** $\pm$ 0.5 | 94.32 $\pm$ 0.2 | **7.28** $\pm$ 1.1 | **0.80** $\pm$ 0.9 |
| $\kappa = 0.1, \tau = 10^{-9}$ | 94.99 $\pm$ 0.2 | 8.59 $\pm$ 1.1 | 1.17 $\pm$ 0.6 | 94.33 $\pm$ 0.3 | 7.39 $\pm$ 1.1 | 1.12 $\pm$ 1.0 |
| $\kappa = 0.1, \tau = 10^{-10}$ | 95.02 $\pm$ 0.3 | 8.62 $\pm$ 1.1 | 1.43 $\pm$ 0.5 | 94.27 $\pm$ 0.3 | 7.29 $\pm$ 1.1 | 0.89 $\pm$ 1.0 |
| $\kappa = 0.1, \tau = 10^{-11}$ | 94.98 $\pm$ 0.1 | 8.48 $\pm$ 1.1 | 1.32 $\pm$ 1.0 | 94.38 $\pm$ 0.2 | **7.27** $\pm$ 1.0 | 1.14 $\pm$ 1.1 |

In order to examine the effects of hyperparameter selection, the sensitivity analysis for the proposed framework is executed. The results for changing $\kappa$ and $\tau$ values are presented in Tables 3 and 4 for Pokec networks and Recidivism network, respectively. Overall, the results demonstrate that the proposed strategy, FairNorm, typically leads to better fairness measures compared to the natural baseline (M-Norm) within a wide range of hyperparameter selection.

### 5.4 Ablation Study

In order to investigate the influences of different fairness regularizers introduced in this study, we carry over an ablation study. The results of this study are presented in Table 5. Table 5 demonstrates that while the employment of $\mathcal{L}_\mu$ seems to have a greater effect in fairness performance improvement, utilization of both $\mathcal{L}_\mu$ and $\mathcal{L}_\Delta$ (FairNorm) leads to better fairness results compared to the cases where only one of the regularizers is used with ReLU activation. For the case where sigmoid is used as the nonlinear activation, use of both regularizers again results in better $\Delta_{EO}$ values compared to the cases where only one of the regularizers is employed. However, the same cannot be always claimed for $\Delta_{SP}$, which can be explained by the fact that sigmoid already limits the maximal deviation to some extend after the first layer reducing the effect of $\mathcal{L}_\Delta$. Overall, Table 5 shows that the utilization of both $\mathcal{L}_\mu$ and $\mathcal{L}_\Delta$ generally achieves the best fairness performance, while $\mathcal{L}_\mu$ appears to be a more influential component compared to $\mathcal{L}_\Delta$ for the bias reduction.

Table 5: Ablation study for the proposed regularizers

| **Pokec-z** | ReLU | | | Sigmoid | | |
|---|---|---|---|---|---|---|
| | Accuracy (%) | $\Delta_{SP}$ (%) | $\Delta_{EO}$ (%) | Accuracy (%) | $\Delta_{SP}$ (%) | $\Delta_{EO}$ (%) |
| M-Norm | $70.71 \pm 0.8$ | $5.57 \pm 1.3$ | $5.00 \pm 2.0$ | $69.84 \pm 0.7$ | $6.21 \pm 1.4$ | $4.44 \pm 2.1$ |
| $\mathcal{L}_{\mu}$ | $70.65 \pm 0.8$ | $1.50 \pm 1.0$ | $2.19 \pm 1.3$ | $69.71 \pm 0.9$ | $\mathbf{1.36} \pm 0.4$ | $1.72 \pm 1.1$ |
| $\mathcal{L}_{\Delta}$ | $70.62 \pm 0.9$ | $5.15 \pm 1.4$ | $4.55 \pm 2.2$ | $69.73 \pm 0.5$ | $5.67 \pm 2.0$ | $4.29 \pm 2.3$ |
| FairNorm | $70.67 \pm 1.0$ | $\mathbf{1.35} \pm 1.2$ | $1.90 \pm 1.8$ | $69.73 \pm 0.9$ | $1.71 \pm 0.3$ | $\mathbf{1.48} \pm 1.1$ |

| **Pokec-n** | ReLU | | | Sigmoid | | |
|---|---|---|---|---|---|---|
| | Accuracy (%) | $\Delta_{SP}$ (%) | $\Delta_{EO}$ (%) | Accuracy (%) | $\Delta_{SP}$ (%) | $\Delta_{EO}$ (%) |
| M-Norm | $69.25 \pm 0.5$ | $2.48 \pm 1.2$ | $2.91 \pm 1.7$ | $68.59 \pm 0.9$ | $1.78 \pm 2.1$ | $2.88 \pm 1.8$ |
| $\mathcal{L}_{\mu}$ | $69.30 \pm 0.7$ | $1.47 \pm 1.2$ | $1.51 \pm 1.4$ | $68.82 \pm 1.1$ | $1.65 \pm 1.2$ | $1.80 \pm 1.8$ |
| $\mathcal{L}_{\Delta}$ | $69.40 \pm 0.5$ | $2.55 \pm 1.1$ | $2.54 \pm 1.6$ | $68.59 \pm 0.9$ | $1.78 \pm 2.1$ | $2.88 \pm 1.8$ |
| FairNorm | $69.38 \pm 0.7$ | $\mathbf{1.26} \pm 1.2$ | $\mathbf{1.22} \pm 1.3$ | $68.88 \pm 1.1$ | $\mathbf{1.44} \pm 1.2$ | $\mathbf{1.74} \pm 1.7$ |

| **Recidivism** | ReLU | | | Sigmoid | | |
|---|---|---|---|---|---|---|
| | Accuracy (%) | $\Delta_{SP}$ (%) | $\Delta_{EO}$ (%) | Accuracy (%) | $\Delta_{SP}$ (%) | $\Delta_{EO}$ (%) |
| M-Norm | $95.00 \pm 0.3$ | $8.87 \pm 1.2$ | $1.71 \pm 0.7$ | $94.45 \pm 0.3$ | $8.94 \pm 1.2$ | $2.06 \pm 1.0$ |
| $\mathcal{L}_{\mu}$ | $95.08 \pm 0.2$ | $8.67 \pm 1.0$ | $1.35 \pm 0.5$ | $94.33 \pm 0.2$ | $\mathbf{7.18} \pm 1.2$ | $1.05 \pm 1.1$ |
| $\mathcal{L}_{\Delta}$ | $95.12 \pm 0.2$ | $8.97 \pm 1.1$ | $1.42 \pm 0.6$ | $94.39 \pm 0.3$ | $8.81 \pm 1.1$ | $1.92 \pm 0.9$ |
| FairNorm | $95.11 \pm 0.2$ | $\mathbf{8.45} \pm 1.0$ | $\mathbf{0.90} \pm 0.5$ | $94.32 \pm 0.2$ | $7.28 \pm 1.1$ | $\mathbf{0.80} \pm 0.9$ |

## 6 Conclusions and Limitations

This study proposes a unified framework, FairNorm, that mitigates bias in GNN-based learning and provides faster convergence in training. FairNorm applies group-wise normalizations, and employs two novel fairness regularizers that manipulate the parameters of these normalizations. The designs of these regularizers are based on theoretical fairness analyses on GNNs. Experimental results on real-world networks show the fairness improvement of FairNorm over fairness-aware baselines in terms of statistical parity and equal opportunity, as well as its similar utility performance in node classification. Furthermore, it is demonstrated that FairNorm improves the convergence speed of the naive baseline where no normalization is used.

The present framework considers only a single sensitive attribute in its design of normalization layers and fairness regularizers. One possible future direction of this study is the extension of the design to a case with multiple sensitive attributes, which may be essential in certain applications.

## 7 Acknowledgements

We thank the anonymous reviewers for their valuable time in helping us improve our manuscript.

Part of this work is supported by the Google Research Scholar Award and NSF ECCS 2207457.

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

# A  Proof of Lemma 1

First, we present the following Lemma, as it will be utilized in the proof of Lemma 1.

**Lemma 2** *(Cauchy Interlace Theorem, (Horn & Johnson, 2012)). Let $\mathbf{A}$ be a Hermitian matrix of order $N$, and let $\mathbf{B}$ be a principal submatrix of $\mathbf{A}$ of order $N-1$, such that $\mathbf{A} = \begin{pmatrix} \mathbf{B} & \mathbf{y} \\ \mathbf{y}^\top & a \end{pmatrix} \in \mathbb{R}^{N \times N}$. If $\lambda_N \leqslant \lambda_{N-1} \leqslant \cdots \leqslant \lambda_2 \leqslant \lambda_1$ lists the eigenvalues of $\mathbf{A}$ and $\gamma_N \leqslant \gamma_{N-1} \leqslant \cdots \leqslant \gamma_3 \leqslant \gamma_2$ the eigenvalues of $\mathbf{B}$, then:*

$$\lambda_N \leqslant \gamma_N \leqslant \lambda_{N-1} \leqslant \gamma_{N-1} \leqslant \cdots \leqslant \lambda_2 \leqslant \gamma_2 \leqslant \lambda_1 \tag{10}$$

*where $\lambda_i = \gamma_i$ only when there is a nonzero $\mathbf{z} \in \mathbb{R}^{N-1}$ such that $\mathbf{Bz} = \gamma_i \mathbf{z}$ and $\mathbf{y}^\top \mathbf{z} = 0$; if $\lambda_i = \gamma_{i-1}$ then there is a nonzero $\mathbf{z} \in \mathbb{R}^{N-1}$ such that $\mathbf{Bz} = \gamma_{i-1}\mathbf{z}, \mathbf{y}^\top \mathbf{z} = 0$.*

The shift operations over different sensitive groups are defined to be:

$$\text{MShift}(\mathbf{W}^{(k)}\mathbf{H}^{(k-1)}\mathbf{Q}) = \mathbf{W}^{(k)}\mathbf{H}^{(k-1)}\mathbf{Q}\mathbf{N}^{(0)}\mathbf{N}^{(1)}, \tag{11}$$

where $\mathbf{N}^{(n)} = \mathbf{I}_N - \frac{1}{|S^n|}\mathbf{e}^{(n)}(\mathbf{e}^{(n)})^\top$ for $n = 0, 1$. Let $0 \leqslant \lambda_1 \leqslant \cdots \leqslant \lambda_N$ be the singular values of $\mathbf{Q}$. Then, eigenvalues of $(\mathbf{Q})^\top \mathbf{Q}$ are $0 \leqslant \lambda_1^2 \leqslant \cdots \leqslant \lambda_N^2$. Let $\gamma_i^2$ denote the eigenvalues of $(\mathbf{N}^{(1)})^\top(\mathbf{N}^{(0)})^\top(\mathbf{Q})^\top \mathbf{Q}\mathbf{N}^{(0)}\mathbf{N}^{(1)} = \mathbf{N}^{(1)}\mathbf{N}^{(0)}(\mathbf{Q})^\top \mathbf{Q}\mathbf{N}^{(0)}\mathbf{N}^{(1)} \ \forall i = 1, \ldots, N$.

$(\mathbf{N}^{(0)}\mathbf{N}^{(1)})$ is a projection matrix, for which the following holds:

$$(\mathbf{N}^{(0)}\mathbf{N}^{(1)})^2 = \mathbf{N}^{(0)}\mathbf{N}^{(1)}\mathbf{N}^{(0)}\mathbf{N}^{(1)} = \mathbf{N}^{(0)}\mathbf{N}^{(0)}\mathbf{N}^{(1)}\mathbf{N}^{(1)} = \mathbf{N}^{(0)}\mathbf{N}^{(1)}, \tag{12}$$

as both $\mathbf{N}^{(0)}$ and $\mathbf{N}^{(1)}$ are symmetric projection matrices onto the orthogonal complement spaces of the subspaces spanned by $\mathbf{e}^{(0)}$ and $\mathbf{e}^{(1)}$, respectively, and $(\mathbf{N}^{(0)}\mathbf{N}^{(1)})$ commutes. Then, the following decomposition can be written: $(\mathbf{N}^{(0)}\mathbf{N}^{(1)}) = \mathbf{U}\,\text{diag}(1, 1, \ldots, 1, 0, 0)\mathbf{U}^\top$, where $\mathbf{U} = \left[\mathbf{U}_{sub}, \frac{1}{\sqrt{|\mathcal{S}^0|}}\mathbf{e}^{(0)}, \frac{1}{\sqrt{|\mathcal{S}^1|}}\mathbf{e}^{(1)}\right]$. Note that $\text{diag}(\mathbf{a})$ creates a diagonal matrix $\mathbf{D} \in \mathbb{R}^{N \times N}$ with $i$th diagonal entry being equal to $a_i, \forall \mathbf{a} \in \mathbb{R}^N$. This decomposition implies that the eigenvalues corresponding to the

eigenvectors $\frac{1}{\sqrt{|\mathcal{S}^0|}}\mathbf{e}^{(0)}, \frac{1}{\sqrt{|\mathcal{S}^1|}}\mathbf{e}^{(1)}$ are zero, which can be shown as:

$$
\begin{aligned}
\mathbf{N}^{(0)}\mathbf{N}^{(1)}\mathbf{e}^{(0)} &= (\mathbf{I}_N - \frac{1}{|S^0|}\mathbf{e}^{(0)}(\mathbf{e}^{(0)})^\top - \frac{1}{|S^1|}\mathbf{e}^{(1)}(\mathbf{e}^{(1)})^\top + \frac{1}{|S^0|}\frac{1}{|S^1|}\mathbf{e}^{(0)}(\mathbf{e}^{(0)})^\top\mathbf{e}^{(1)}(\mathbf{e}^{(1)})^\top)\mathbf{e}^{(0)} \\
&= (\mathbf{I}_N - \frac{1}{|S^0|}\mathbf{e}^{(0)}(\mathbf{e}^{(0)})^\top - \frac{1}{|S^1|}\mathbf{e}^{(1)}(\mathbf{e}^{(1)})^\top)\mathbf{e}^{(0)} \text{ , as } \mathbf{e}^{(0)} \text{ and } \mathbf{e}^{(1)} \text{ are orthogonal,} \\
&= \mathbf{e}^{(0)} - \frac{1}{|S^0|}\mathbf{e}^{(0)}(\mathbf{e}^{(0)})^\top\mathbf{e}^{(0)} - \frac{1}{|S^1|}\mathbf{e}^{(1)}(\mathbf{e}^{(1)})^\top\mathbf{e}^{(0)} \\
&= \mathbf{e}^{(0)} - \frac{1}{|S^0|}|S^0|\mathbf{e}^{(0)} \text{ , as } (\mathbf{e}^{(0)})^\top\mathbf{e}^{(0)} = |S^0| \text{ by definition,} \\
&= 0\mathbf{e}^{(0)}.
\end{aligned}
\tag{13}
$$

The same analysis also holds for $\mathbf{e}^{(1)}$. Let $\gamma_N^2$ and $\gamma_{N-1}^2$ denote zero eigenvalues, and $0 \leqslant \gamma_1^2 \leqslant \cdots \leqslant \gamma_{N-2}^2$ hold. Based on the decomposition of $(\mathbf{N}^{(0)}\mathbf{N}^{(1)})$, the following can be written:

$$
\begin{aligned}
(\mathbf{N}^{(1)})^\top(\mathbf{N}^{(0)})^\top(\mathbf{Q})^\top\mathbf{Q}\mathbf{N}^{(0)}\mathbf{N}^{(1)} &= \mathbf{N}^{(1)}\mathbf{N}^{(0)}(\mathbf{Q})^\top\mathbf{Q}\mathbf{N}^{(0)}\mathbf{N}^{(1)} \\
&= \mathbf{U}\operatorname{diag}(1,1,\ldots,1,0,0)\mathbf{U}^\top(\mathbf{Q})^\top\mathbf{Q}\mathbf{U}\operatorname{diag}(1,1,\ldots,1,0,0)\mathbf{U}^\top \\
&\sim \operatorname{diag}(1,1,\ldots,1,0,0)\mathbf{U}^\top(\mathbf{Q})^\top\mathbf{Q}\mathbf{U}\operatorname{diag}(1,1,\ldots,1,0,0),
\end{aligned}
\tag{14}
$$

where $\mathbf{A} \sim \mathbf{B}$, if the eigenvalues of $\mathbf{A}$ and $\mathbf{B}$ are the same. Furthermore, denote $\operatorname{diag}(1,1,\ldots,1,0,0)$ by $\mathbf{D} \in \mathbb{R}^{N\times N}$:

$$
\mathbf{D} = \begin{bmatrix} \mathbf{I}_{N-2} & [\mathbf{0}]_{N-2\times 2} \\ [\mathbf{0}]_{2\times N-2} & [\mathbf{0}]_{2\times 2} \end{bmatrix},
\tag{15}
$$

where $[\mathbf{0}]_{i\times j}$ denotes an all-zeros matrix with dimensions $i \times j$. Let $\mathbf{C}$ denote $\mathbf{Q}^\top\mathbf{Q}$.

$$
\begin{aligned}
&(\mathbf{N}^{(1)})^\top(\mathbf{N}^{(0)})^\top\mathbf{C}\mathbf{N}^{(0)}\mathbf{N}^{(1)} \\
&\sim \mathbf{D}\mathbf{U}^\top\mathbf{C}\mathbf{U}\mathbf{D} \\
&= \mathbf{D}\begin{bmatrix} \mathbf{U}_{sub}^\top \\ \frac{1}{\sqrt{|\mathcal{S}^0|}}(\mathbf{e}^{(0)})^\top \\ \frac{1}{\sqrt{|\mathcal{S}^1|}}(\mathbf{e}^{(1)})^\top \end{bmatrix}\mathbf{C}\begin{bmatrix} \mathbf{U}_{sub} & \frac{1}{\sqrt{|\mathcal{S}^0|}}(\mathbf{e}^{(0)}) & \frac{1}{\sqrt{|\mathcal{S}^1|}}(\mathbf{e}^{(1)}) \end{bmatrix}\mathbf{D} \\
&= \mathbf{D}\begin{bmatrix} \mathbf{U}_{sub}^\top\mathbf{C}\mathbf{U}_{sub} & \frac{1}{\sqrt{|\mathcal{S}^0|}}\mathbf{U}_{sub}^\top\mathbf{C}(\mathbf{e}^{(0)}) & \frac{1}{\sqrt{|\mathcal{S}^1|}}\mathbf{U}_{sub}^\top\mathbf{C}(\mathbf{e}^{(1)}) \\ \frac{1}{\sqrt{|\mathcal{S}^0|}}(\mathbf{e}^{(0)})^\top\mathbf{C}\mathbf{U}_{sub} & \frac{1}{|\mathcal{S}^0|}(\mathbf{e}^{(0)})^\top\mathbf{C}\mathbf{e}^{(0)} & \frac{1}{\sqrt{|\mathcal{S}^0||\mathcal{S}^1|}}(\mathbf{e}^{(0)})^\top\mathbf{C}\mathbf{e}^{(1)} \\ \frac{1}{\sqrt{|\mathcal{S}^1|}}(\mathbf{e}^{(1)})^\top\mathbf{C}\mathbf{U}_{sub} & \frac{1}{\sqrt{|\mathcal{S}^0||\mathcal{S}^1|}}(\mathbf{e}^{(1)})^\top\mathbf{C}\mathbf{e}^{(0)} & \frac{1}{|\mathcal{S}^1|}(\mathbf{e}^{(1)})^\top\mathbf{C}\mathbf{e}^{(1)} \end{bmatrix}\mathbf{D} \\
&= \begin{bmatrix} \mathbf{U}_{sub}^\top\mathbf{C}\mathbf{U}_{sub} & [\mathbf{0}]_{N-2\times 2} \\ [\mathbf{0}]_{2\times N-2} & [\mathbf{0}]_{2\times 2} \end{bmatrix}.
\end{aligned}
\tag{16}
$$

As, $(\mathbf{N}^{(1)})^\top(\mathbf{N}^{(0)})^\top\mathbf{C}\mathbf{N}^{(0)}\mathbf{N}^{(1)}$ has the eigenvalues $0 \leqslant \gamma_1^2 \leqslant \cdots \leqslant \gamma_{N-2}^2$, and $\gamma_{N-1}^2 = \gamma_N^2 = 0$, equation 16 shows that $\mathbf{U}_{sub}^\top\mathbf{C}\mathbf{U}_{sub} \in \mathbb{R}^{N-2\times N-2}$ has the eigenvalues $0 \leqslant \gamma_1^2 \leqslant \cdots \leqslant \gamma_{N-2}^2$.

Let $\mathbf{R}$ denote $\mathbf{U}^\top\mathbf{C}\mathbf{U}$, then $\mathbf{R}$ has the eigenvalues $0 \leqslant \lambda_1^2 \leqslant \cdots \leqslant \lambda_N^2$, as $\mathbf{R} \sim \mathbf{C}$.

$$
\mathbf{R} = \begin{bmatrix} \mathbf{U}_{sub}^\top\mathbf{C}\mathbf{U}_{sub} & \frac{1}{\sqrt{|\mathcal{S}^0|}}\mathbf{U}_{sub}^\top\mathbf{C}(\mathbf{e}^{(0)}) & \frac{1}{\sqrt{|\mathcal{S}^1|}}\mathbf{U}_{sub}^\top\mathbf{C}(\mathbf{e}^{(1)}) \\ \frac{1}{\sqrt{|\mathcal{S}^0|}}(\mathbf{e}^{(0)})^\top\mathbf{C}\mathbf{U}_{sub} & \frac{1}{|\mathcal{S}^0|}(\mathbf{e}^{(0)})^\top\mathbf{C}\mathbf{e}^{(0)} & \frac{1}{\sqrt{|\mathcal{S}^0||\mathcal{S}^1|}}(\mathbf{e}^{(0)})^\top\mathbf{C}\mathbf{e}^{(1)} \\ \frac{1}{\sqrt{|\mathcal{S}^1|}}(\mathbf{e}^{(1)})^\top\mathbf{C}\mathbf{U}_{sub} & \frac{1}{\sqrt{|\mathcal{S}^0||\mathcal{S}^1|}}(\mathbf{e}^{(1)})^\top\mathbf{C}\mathbf{e}^{(0)} & \frac{1}{|\mathcal{S}^1|}(\mathbf{e}^{(1)})^\top\mathbf{C}\mathbf{e}^{(1)} \end{bmatrix}
\tag{17}
$$

Further, define matrix $\mathbf{A} \in \mathbb{R}^{N-1\times N-1}$ such that:

$$
\mathbf{A} = \begin{bmatrix} \mathbf{U}_{sub}^\top\mathbf{C}\mathbf{U}_{sub} & \frac{1}{\sqrt{|\mathcal{S}^0|}}\mathbf{U}_{sub}^\top\mathbf{C}(\mathbf{e}^{(0)}) \\ \frac{1}{\sqrt{|\mathcal{S}^0|}}(\mathbf{e}^{(0)})^\top\mathbf{C}\mathbf{U}_{sub} & \frac{1}{|\mathcal{S}^0|}(\mathbf{e}^{(0)})^\top\mathbf{C}\mathbf{e}^{(0)} \end{bmatrix},
\tag{18}
$$

together with eigenvalues $\eta_1 \leqslant \cdots \leqslant \eta_{N-1}$. Then, utilizing Lemma 2, equation 17 and equation 18, we can conclude that

$$\lambda_1^2 \leqslant \eta_1 \leqslant \lambda_2^2 \leqslant \eta_2 \cdots \leqslant \eta_{N-1} \leqslant \lambda_N^2, \tag{19}$$

where $\lambda_i^2 = \eta_i$ or $\lambda_i^2 = \eta_{i-1}$, if and only if there is a right singular vector $\boldsymbol{\alpha}$ of $\mathbf{Q}$ such that $(\mathbf{e}^{(1)})^\top \boldsymbol{\alpha} = 0$. The proof of the condition for $\lambda_i^2 = \eta_i$ or $\lambda_i^2 = \eta_{i-1}$ is presented below in *italic*.

**Proof:** *Cauchy interlace theorem states that inequalities in equation 10 become equalities if there is a nonzero $\mathbf{z} \in \mathbb{R}^{N-1}$ such that $\mathbf{Bz} = \gamma_i \mathbf{z}$ and $\mathbf{y}^\top \mathbf{z} = 0$ or if there is a nonzero $\mathbf{z} \in \mathbb{R}^{N-1}$ such that $\mathbf{Bz} = \gamma_{i-1} \mathbf{z}$ and $\mathbf{y}^\top \mathbf{z} = 0$. Therefore, for the result in equation 19, inequalities become equalities, if there is a nonzero $\mathbf{z} \in \mathbb{R}^{N-1}$ such that:*

$$\mathbf{Az} = \eta \mathbf{z} \; and \; \left[ \; \frac{1}{\sqrt{|\mathcal{S}^1|}} (\mathbf{e}^{(1)})^\top \mathbf{CU}_{sub} \quad \frac{1}{\sqrt{|\mathcal{S}^0||\mathcal{S}^1|}} (\mathbf{e}^{(1)})^\top \mathbf{Ce}^{(0)} \; \right] \mathbf{z} = 0. \tag{20}$$

*Note that we dropped the subscript of $\eta$ in equation 20, as it is enough to hold these conditions for any of the $\eta_i s$ to turn one of the inequalities into equality in equation 19.*

$$\mathbf{A} = \left[ \begin{array}{c} \mathbf{U}_{sub}^\top \\ \frac{1}{\sqrt{|\mathcal{S}^0|}} (\mathbf{e}^{(0)})^\top \end{array} \right] \mathbf{C} \left[ \; \mathbf{U}_{sub} \quad \frac{1}{\sqrt{|\mathcal{S}^0|}} \mathbf{e}^{(0)} \; \right] \tag{21}$$

*Let $\mathbf{U}_1 := \left[ \; \mathbf{U}_{sub} \quad \frac{1}{\sqrt{|\mathcal{S}^0|}} \mathbf{e}^{(0)} \; \right]$, where $\mathbf{U}_1$ forms an orthogonal basis for $\mathbf{e}^{(1)}$. Therefore, $\mathbf{A} = \mathbf{U}_1^\top \mathbf{CU}_1$. The conditions presented in equation 20 can be rewritten based on this definition:*

$$\mathbf{U}_1^\top \mathbf{CU}_1 \mathbf{z} = \eta \mathbf{z} \; and \; \frac{1}{\sqrt{|\mathcal{S}^1|}} (\mathbf{e}^{(1)})^\top \mathbf{CU}_1 \mathbf{z} = 0. \tag{22}$$

*The second condition in equation 22 demonstrates that for inequalities in equation 19 become equalities, $\mathbf{CU}_1 \mathbf{z}$ should lie in the orthogonal complement space of $\mathbf{e}^{(1)}$, which is spanned by $\mathbf{U}_1$. Therefore, if the second condition in equation 22 is satisfied, there exists a vector $\mathbf{b} \in \mathbb{R}^{N-1}$ such that:*

$$\mathbf{CU}_1 \mathbf{z} = \mathbf{U}_1 \mathbf{b}. \tag{23}$$

*In this case, the first condition in equation 22 becomes:*

$$\mathbf{U}_1^\top \mathbf{CU}_1 \mathbf{z} = \mathbf{U}_1^\top \mathbf{U}_1 \mathbf{b} = \mathbf{I}_{N-1} \mathbf{b} = \mathbf{b} = \eta \mathbf{z} \tag{24}$$

*Therefore, the following equality should hold to meet both criterion in equation 20:*

$$\mathbf{CU}_1 \mathbf{z} = \mathbf{U}_1 \mathbf{b} = \eta \mathbf{U}_1 \mathbf{z}. \tag{25}$$

*equation 25 demonstrates that the conditions in equation 20 are met, if $\mathbf{U}_1 \mathbf{z}$ is the eigenvector of $\mathbf{C} = \mathbf{Q}^\top \mathbf{Q}$ associated with eigenvalue $\eta$. This eigenvector $\mathbf{U}_1 \mathbf{z}$ lies in the orthogonal complement space of $\mathbf{e}^{(0)}$ and the eigenvector of $\mathbf{C} = \mathbf{Q}^\top \mathbf{Q}$ is the right singular vector of $\mathbf{Q}$. Therefore, inequalities in equation 19 become equalities, if there is a right singular vector $\boldsymbol{\alpha}$ of $\mathbf{Q}$ such that $(\mathbf{e}^{(1)})^\top \boldsymbol{\alpha} = 0$, which concludes the proof.*

$\mathbf{A}$ is created in the following way:

$$\mathbf{A} = \left[ \begin{array}{cc} \mathbf{U}_{sub}^\top \mathbf{CU}_{sub} & \frac{1}{\sqrt{|\mathcal{S}^0|}} \mathbf{U}_{sub}^\top \mathbf{C}(\mathbf{e}^{(0)}) \\ \frac{1}{\sqrt{|\mathcal{S}^0|}} (\mathbf{e}^{(0)})^\top \mathbf{CU}_{sub} & \frac{1}{|\mathcal{S}^0|} (\mathbf{e}^{(0)})^\top \mathbf{Ce}^{(0)} \end{array} \right], \tag{26}$$

together with eigenvalues $\eta_1 \leqslant \cdots \leqslant \eta_{N-1}$. Furthermore, equation 16 shows that $\mathbf{U}_{sub}^\top \mathbf{CU}_{sub} \in \mathbb{R}^{N-2 \times N-2}$ has the eigenvalues $0 \leqslant \gamma_1^2 \leqslant \cdots \leqslant \gamma_{N-2}^2$. Again, we can apply Cauchy interlace theorem presented in Lemma 2, which concludes that:

$$\eta_1 \leqslant \gamma_1^2 \leqslant \eta_2 \leqslant \gamma_2^2 \cdots \leqslant \gamma_{N-2}^2 \leqslant \eta_{N-1}, \tag{27}$$

where $\eta_i = \gamma_i^2$ or $\eta_i = \gamma_{i-1}^2$, if and only if there is a right singular vector $\boldsymbol{\alpha}$ of $\mathbf{Q}$ such that $(\mathbf{e}^{(0)})^\top \boldsymbol{\alpha} = 0$ and $(\mathbf{e}^{(1)})^\top \boldsymbol{\alpha} = 0$. For these conditions leading to equalities $\eta_i = \gamma_i^2$ or $\eta_i = \gamma_{i-1}^2$, the proof follows in the same manner as the previous one.

Finally, by unifying the results of equation 19 and equation 27, the Theorem 1 can be proved, such that:

$$\lambda_1 \leqslant \gamma_1 \tag{28}$$

$$\lambda_2 \leqslant \gamma_2 \tag{29}$$

$$\vdots$$

$$\lambda_{N-2} \leqslant \gamma_{N-2} \leqslant \lambda_N \tag{30}$$

where $\lambda_i = \gamma_i$ or $\lambda_N = \gamma_{N-2}$, only if $\mathbf{Q}$ have a right singular vector $\boldsymbol{\alpha}$ such that $(\mathbf{e}^{(0)})^\top \boldsymbol{\alpha} = 0$ and $(\mathbf{e}^{(1)})^\top \boldsymbol{\alpha} = 0$. Note that $\lambda_i$s and $\gamma_i$s are defined to be non-negative, thus we can omit the powers of 2 in the final result.

## B    Learning Environment and Proof for Theorem 1

We first introduce the following Lemma that will be utilized in the proof.

**Lemma 3** *(Theorem 1, (Cui et al., 2019)) Let $\boldsymbol{x}_1, \ldots, \boldsymbol{x}_N \sim \mathcal{N}(\mathbf{0}, \boldsymbol{\Sigma})$ be independent Gaussian vectors, where $\boldsymbol{\Sigma}$ is an $F \times F$ real positive definite matrix. Let $\boldsymbol{B}$ be a fixed symmetric real $N \times N$ matrix. Consider the compound Wishart matrix $\boldsymbol{W} = \boldsymbol{X}\boldsymbol{B}\boldsymbol{X}^T/N$ with $\boldsymbol{X} = [\boldsymbol{x}_1, \ldots, \boldsymbol{x}_N]$. Then for any $\delta \geqslant 0$, the following event*

$$\|\boldsymbol{W} - \mathbb{E}\boldsymbol{W}\|_2 \geqslant \frac{32\|\boldsymbol{B}\|_F \delta + 64\|\boldsymbol{B}\|_2 \delta^2}{N}\|\boldsymbol{\Sigma}\|_2$$

*holds with probability at most $2\exp\left(-2\delta^2 + 2F\log 3\right)$, where $\|\cdot\|_F$ denotes the Frobenius norm. Specifically, for $\delta \geqslant \sqrt{2\ln(3)}\sqrt{F}$,*

$$\|\boldsymbol{W} - \mathbb{E}\boldsymbol{W}\|_2 \geqslant \frac{32\|\boldsymbol{B}\|_F \delta + 64\|\boldsymbol{B}\|_2 \delta^2}{N}\|\boldsymbol{\Sigma}\|_2$$

*holds with probability at most $2\exp\left(-\delta^2\right)$.*

**Model:**    For the node classification task over a graph $\mathcal{G}$, a linear one-layer GNN-based model is considered, where the models without and with individual shifts for different sensitive groups are denoted by $f^{vanilla}(\mathbf{X}, \mathbf{Q}) = \mathbf{w}^\top \mathbf{X}\mathbf{Q}$ and $f^{MShift}(\mathbf{X}, \mathbf{Q}) = \mathbf{w}^\top \mathbf{X}\mathbf{Q}\mathbf{N}^{(0)}\mathbf{N}^{(1)}$, respectively for $\mathbf{w} \in \mathbb{R}^{F \times 1}$, $\mathbf{X} \in \mathbb{R}^{F \times N}$ and $\mathbf{Q} \in \mathbb{R}^{N \times N}$.

**Training loss:**    Let $\mathbf{Z}^{vanilla} \in \mathbb{R}^{F \times N}$ and $\mathbf{Z}^{MShift} \in \mathbb{R}^{F \times N}$ denote $\mathbf{X}\mathbf{Q}$ and $\mathbf{X}\mathbf{Q}\mathbf{N}^{(0)}\mathbf{N}^{(1)}$, respectively. The training loss for both the models without and with individual shifts for different sensitive groups follows as:

$$\mathcal{L}(\mathbf{w}) = \frac{1}{2}\|\mathbf{Z}^\top \mathbf{w} - \mathbf{y}\|_2^2 \tag{31}$$

where $\mathbf{y} \in \mathbb{R}^N$ denotes the labels of the nodes. Note that this loss is applicable to both $\mathbf{Z}^{vanilla}$ and $\mathbf{Z}^{MShift}$, and $\mathbf{Z}$ is used interchangeably with $\mathbf{Z}^{vanilla}$ for the ease of notation.

**Optimization:**    Gradient descent is utilized for the optimization by initializing $\mathbf{w}_0 = \mathbf{0}$, where an optimization step can be written as:

$$\begin{aligned}
\mathbf{w}_{t+1} &= \mathbf{w}_t - \eta(\mathbf{Z}\mathbf{Z}^\top \mathbf{w}_t - \mathbf{Z}\mathbf{y}) \\
&= (\mathbf{I}_F - \eta\mathbf{Z}\mathbf{Z}^\top)\mathbf{w}_t + \eta\mathbf{Z}\mathbf{y}.
\end{aligned} \tag{32}$$

Note that $\eta$ in Equation equation 32 denotes the learning rate.

In this setting, $\mathbf{w}_t$ converges to optimal solution $\mathbf{w}^* = (\mathbf{Z}\mathbf{Z}^\top)^\dagger \mathbf{Z}\mathbf{y}$ according to the solution of least squares problem (Horn & Johnson, 2012), where superscript $\dagger$ denotes Moore-Penrose inverse. The residual of $\mathbf{w}_{t+1}$ follows as:

$$\begin{aligned}
\mathbf{w}_{t+1} - \mathbf{w}^* &= (\mathbf{I}_F - \eta\mathbf{Z}\mathbf{Z}^\top)\mathbf{w}_t + \eta\mathbf{Z}\mathbf{y} - \mathbf{w}^* \\
&= (\mathbf{I}_F - \eta\mathbf{Z}\mathbf{Z}^\top)\mathbf{w}_t + \eta(\mathbf{Z}\mathbf{Z}^\top)(\mathbf{Z}\mathbf{Z}^\top)^\dagger \mathbf{Z}\mathbf{y} - \mathbf{w}^* \\
&= (\mathbf{I}_F - \eta\mathbf{Z}\mathbf{Z}^\top)\mathbf{w}_t - (\mathbf{I}_F - \eta\mathbf{Z}\mathbf{Z}^\top)\mathbf{w}^* \text{ , as } \mathbf{w}^* = (\mathbf{Z}\mathbf{Z}^\top)^\dagger \mathbf{Z}\mathbf{y} \\
&= (\mathbf{I}_F - \eta\mathbf{Z}\mathbf{Z}^\top)(\mathbf{w}_t - \mathbf{w}^*).
\end{aligned} \tag{33}$$

**Assumption 1:** $\mathbf{w}_0 = \mathbf{0}$.

Based on Assumption 1 and equation 33, following inequality can be written:

$$\|\mathbf{w}_t - \mathbf{w^*}\| \leqslant \|(\mathbf{I}_F - \eta \mathbf{Z}\mathbf{Z}^\top)\|^t \|\mathbf{w^*}\|. \tag{34}$$

For the learning rate $\eta = \frac{1}{\sigma_{max}(\mathbf{Z}\mathbf{Z}^\top)}$, the following convergence rate can be guaranteed for this problem:

$$\|\mathbf{w}_t - \mathbf{w^*}\| \leqslant \left(1 - \frac{\sigma_{min}(\mathbf{Z}\mathbf{Z}^\top)}{\sigma_{max}(\mathbf{Z}\mathbf{Z}^\top)}\right)^t \|\mathbf{w^*}\|. \tag{35}$$

Note that $\sigma_{min}$ and $\sigma_{max}$ output the minimum and maximum positive eigenvalues of the input matrix, respectively. The same result can also be derived for $\mathbf{Z}^{MShift}$ following the same steps, such as:

$$\|\mathbf{w}_t - \mathbf{w^*}\| \leqslant \left(1 - \frac{\sigma_{min}(\mathbf{Z}^{MShift}(\mathbf{Z}^{MShift})^\top)}{\sigma_{max}(\mathbf{Z}^{MShift}(\mathbf{Z}^{MShift})^\top)}\right)^t \|\mathbf{w^*}\|. \tag{36}$$

Equations equation 35 and equation 36 demonstrate that the convergence rates of the defined node classification problem for without and with shifts depend on the terms $1 - \left(\frac{\sigma_{min}(\mathbf{Z}\mathbf{Z}^\top)}{\sigma_{max}(\mathbf{Z}\mathbf{Z}^\top)}\right)$ and $1 - \left(\frac{\sigma_{min}(\mathbf{Z}^{MShift}(\mathbf{Z}^{MShift})^\top)}{\sigma_{max}(\mathbf{Z}^{MShift}(\mathbf{Z}^{MShift})^\top)}\right)$, respectively. Therefore, the next step examines these factors together with the following assumptions.

**Assumption 2:** $\mathbf{x} \in \mathbb{R}^F$ **is a centered Gaussian random variable with covariance matrix** $\boldsymbol{\Sigma} = \mathrm{E}_\mathbf{x}[\mathbf{x}\mathbf{x}^\top]$. **The features of nodes** $\mathbf{x}_1, \ldots, \mathbf{x}_N$ **are independent realizations of** $\mathbf{x}$, **where** $\mathbf{X} = [\mathbf{x}_1 \cdots \mathbf{x}_N]$. $\mathbf{Z} = \mathbf{X}Q$ **and** $\mathbf{Z}^{MShift} = \mathbf{X}Q\mathbf{N}^{(0)}\mathbf{N}^{(1)}$.

**Assumption 3:** $\mathrm{E}[\mathbf{X}\mathbf{Q}] := \mathbf{Y}$.

**Assumption 4:** $\mathbf{O} \leqslant \mathrm{E}[\frac{(\mathbf{X}\mathbf{Q}-\mathbf{Y})}{\sqrt{N}} \frac{(\mathbf{X}\mathbf{Q}-\mathbf{Y})}{\sqrt{N}}^\top] \leqslant \delta_1 \mathbf{I}_F$

**Assumption 5:** $\mathbf{O} \leqslant \mathrm{E}[\frac{(\mathbf{X}\mathbf{Q}-\mathbf{Y})}{\sqrt{N}}\mathbf{N}^{(0)}\mathbf{N}^{(1)} \frac{(\mathbf{X}\mathbf{Q}-\mathbf{Y})}{\sqrt{N}}^\top] \leqslant \delta_1 \mathbf{I}_F$

**Assumption 6: Defined $\mathbf{Y}$ matrix is full rank.**

We will analyze the eigenvalues of $\frac{1}{N}\mathbf{Z}\mathbf{Z}^\top$, as multiplying with a constant does not change the ratio between the minimum and maximum eigenvalues.

$$\frac{1}{N}\mathbf{Z}\mathbf{Z}^\top = \frac{1}{N}\sum_{i=1}^N \mathbf{z}_i\mathbf{z}_i^\top. \tag{37}$$

For the analysis of $\frac{1}{N}\mathbf{Z}\mathbf{Z}^\top$, we will first focus on $\mathrm{E}\left[\frac{\mathbf{Z}\mathbf{Z}^\top}{N}\right]$, which equals to:

$$\begin{aligned}
\mathrm{E}\left[\frac{\mathbf{Z}\mathbf{Z}^\top}{N}\right] &= \mathrm{E}\left[\frac{\mathbf{X}\mathbf{Q}}{\sqrt{N}}\left(\frac{\mathbf{X}\mathbf{Q}}{\sqrt{N}}\right)^\top\right] \\
&= \frac{\mathbf{Y}\mathbf{Y}^\top}{N} + \mathrm{E}[\frac{(\mathbf{X}\mathbf{Q}-\mathbf{Y})}{\sqrt{N}} \frac{(\mathbf{X}\mathbf{Q}-\mathbf{Y})}{\sqrt{N}}^\top]
\end{aligned} \tag{38}$$

based on the Assumption 3. By utilizing Weyl's inequality and the provided assumptions, following inequalities can be written:

$$\begin{aligned}
\sigma_{max}(\frac{\mathbf{Y}\mathbf{Y}^\top}{N}) &\leqslant \sigma_{max}(\mathrm{E}\left[\frac{\mathbf{Z}\mathbf{Z}^\top}{N}\right]) \leqslant \sigma_{max}(\frac{\mathbf{Y}\mathbf{Y}^\top}{N}) + \delta_1 \\
\sigma_{min}(\frac{\mathbf{Y}\mathbf{Y}^\top}{N}) &\leqslant \sigma_{min}(\mathrm{E}\left[\frac{\mathbf{Z}\mathbf{Z}^\top}{N}\right]) \leqslant \sigma_{min}(\frac{\mathbf{Y}\mathbf{Y}^\top}{N}) + \delta_1
\end{aligned} \tag{39}$$

Similarly:

$$
\begin{aligned}
\mathrm{E}\left[\frac{\mathbf{Z}^{MShift}(\mathbf{Z}^{MShift})^\top}{N}\right] &= \mathrm{E}\left[\frac{\mathbf{XQN}^{(0)}\mathbf{N}^{(1)}}{\sqrt{N}}\frac{(\mathbf{XQN}^{(0)}\mathbf{N}^{(1)})^\top}{\sqrt{N}}\right] \\
&= \mathrm{E}\left[\frac{\mathbf{XQ}}{\sqrt{N}}(\mathbf{N}^{(0)}\mathbf{N}^{(1)})^2\frac{(\mathbf{XQ})^\top}{\sqrt{N}}\right] \text{ , as } \mathbf{N}^{(0)}\mathbf{N}^{(1)} \text{ commutes,} \\
&= \mathrm{E}\left[\frac{\mathbf{XQ}}{\sqrt{N}}\mathbf{N}^{(0)}\mathbf{N}^{(1)}\frac{(\mathbf{XQ})^\top}{\sqrt{N}}\right] \text{ , as } \mathbf{N}^{(0)}\mathbf{N}^{(1)} \text{ is a projection matrix} \\
&= \frac{\mathbf{YN}^{(0)}\mathbf{N}^{(1)}\mathbf{Y}^\top}{N} + \mathrm{E}\left[\frac{(\mathbf{XQ}-\mathbf{Y})}{\sqrt{N}}\mathbf{N}^{(0)}\mathbf{N}^{(1)}\frac{(\mathbf{XQ}-\mathbf{Y})^\top}{\sqrt{N}}\right]
\end{aligned}
\tag{40}
$$

Again, by utilizing Weyl's inequality and the provided assumptions, following inequalities can be written for the shifted case:

$$
\begin{aligned}
\sigma_{max}(\frac{\mathbf{YN}^{(0)}\mathbf{N}^{(1)}\mathbf{Y}^\top}{N}) &\leqslant \sigma_{max}(\mathrm{E}\left[\frac{\mathbf{Z}^{MShift}(\mathbf{Z}^{MShift})^\top}{N}\right]) \leqslant \sigma_{max}(\frac{\mathbf{YN}^{(0)}\mathbf{N}^{(1)}\mathbf{Y}^\top}{N}) + \delta_1 \\
\sigma_{min}(\frac{\mathbf{YN}^{(0)}\mathbf{N}^{(1)}\mathbf{Y}^\top}{N}) &\leqslant \sigma_{min}(\mathrm{E}\left[\frac{\mathbf{Z}^{MShift}(\mathbf{Z}MShift)^\top}{N}\right]) \leqslant \sigma_{min}(\frac{\mathbf{YN}^{(0)}\mathbf{N}^{(1)}\mathbf{Y}^\top}{N}) + \delta_1
\end{aligned}
\tag{41}
$$

Next step in the proof is bounding the errors $\|\frac{1}{N}\mathbf{ZZ}^\top - \mathrm{E}\left[\frac{\mathbf{ZZ}^\top}{N}\right]\|_2$ and $\|\frac{1}{N}\mathbf{Z}^{MShift}(\mathbf{Z}^{MShift})^\top - \mathrm{E}\left[\frac{\mathbf{Z}^{MShift}(\mathbf{Z}^{MShift})^\top}{N}\right]\|_2$. Based on Lemma 3 and assumption 2, the following inequality holds with probability $1-\epsilon$, where $\epsilon < \frac{2}{e}$, for errors $\|\frac{1}{N}\mathbf{ZZ}^\top - \mathrm{E}\left[\frac{\mathbf{ZZ}^\top}{N}\right]\|_2$ and $\|\frac{1}{N}\mathbf{Z}^{MShift}(\mathbf{Z}^{MShift})^\top - \mathrm{E}\left[\frac{\mathbf{Z}^{MShift}(\mathbf{Z}^{MShift})^\top}{N}\right]\|_2$:

$$
\left\|\frac{1}{N}\mathbf{ZZ}^\top - \mathrm{E}\left[\frac{\mathbf{ZZ}^\top}{N}\right]\right\|_2 \leqslant \mathcal{O}\left(\frac{log(2/\epsilon)}{N}\right),
\tag{42}
$$

where the constants $\|\mathbf{QQ}^\top\|_F$, $\|\mathbf{QQ}^\top\|_2$, and $\|\mathbf{\Sigma}\|_2$ are hidden in $\mathcal{O}(.)$ notation for a focus on $N$, similar to the approach taken in Cai et al. (2021). Similarly, the following bound holds for $\|\frac{1}{N}\mathbf{Z}^{MShift}(\mathbf{Z}^{MShift})^\top - \mathrm{E}\left[\frac{\mathbf{Z}^{MShift}(\mathbf{Z}^{MShift})^\top}{N}\right]\|_2$ with probability $1-\epsilon$, where $\epsilon < \frac{2}{e}$:

$$
\left\|\frac{1}{N}\mathbf{Z}^{MShift}(\mathbf{Z}^{MShift})^\top - \mathrm{E}\left[\frac{\mathbf{Z}^{MShift}(\mathbf{Z}^{MShift})^\top}{N}\right]\right\|_2 \leqslant \mathcal{O}\left(\frac{log(2/\epsilon)}{N}\right).
\tag{43}
$$

Based on Lemma 3, Equations equation 42 and equation 43, the inequalities in equation 39 and equation 41 can be re-organized. Following inequalities hold with probability $1-\epsilon$ where $\epsilon < \frac{2}{e}$:

$$
\begin{aligned}
\sigma_{max}(\frac{\mathbf{YY}^\top}{N}) - \mathcal{O}\left(\frac{log(2/\epsilon)}{N}\right) &\leqslant \sigma_{max}(\frac{\mathbf{ZZ}^\top}{N}) \leqslant \sigma_{max}(\frac{\mathbf{YY}^\top}{N}) + \delta_1 + \mathcal{O}\left(\frac{log(2/\epsilon)}{N}\right) \\
\sigma_{min}(\frac{\mathbf{YY}^\top}{N}) - \mathcal{O}\left(\frac{log(2/\epsilon)}{N}\right) &\leqslant \sigma_{min}(\frac{\mathbf{ZZ}^\top}{N}) \leqslant \sigma_{min}(\frac{\mathbf{YY}^\top}{N}) + \delta_1 + \mathcal{O}\left(\frac{log(2/\epsilon)}{N}\right)
\end{aligned}
\tag{44}
$$

$$\sigma_{max}(\frac{\mathbf{Y}\mathbf{N}^{(0)}\mathbf{N}^{(1)}\mathbf{Y}^\top}{N}) - \mathcal{O}\left(\frac{log(2/\epsilon)}{N}\right) \leqslant \sigma_{max}(\frac{1}{N}\mathbf{Z}^{MShift}(\mathbf{Z}^{MShift})^\top)$$

$$\leqslant \sigma_{max}(\frac{\mathbf{Y}\mathbf{N}^{(0)}\mathbf{N}^{(1)}\mathbf{Y}^\top}{N}) + \delta_1 + \mathcal{O}\left(\frac{log(2/\epsilon)}{N}\right)$$

$$\sigma_{min}(\frac{\mathbf{Y}\mathbf{N}^{(0)}\mathbf{N}^{(1)}\mathbf{Y}^\top}{N}) - \mathcal{O}\left(\frac{log(2/\epsilon)}{N}\right) \leqslant \sigma_{min}(\frac{1}{N}\mathbf{Z}^{MShift}(\mathbf{Z}^{MShift})^\top)$$

$$\leqslant \sigma_{min}(\frac{\mathbf{Y}\mathbf{N}^{(0)}\mathbf{N}^{(1)}\mathbf{Y}^\top}{N}) + \delta_1 + \mathcal{O}\left(\frac{log(2/\epsilon)}{N}\right)$$
$$(45)$$

By the same method that proves Theorem 1 and is presented in Appendix A, it can be shown that positive eigenvalues of $\mathbf{Y}\mathbf{N}^{(0)}\mathbf{N}^{(1)}\mathbf{Y}^\top$ are interlaced between the positive eigenvalues of $\mathbf{Y}\mathbf{Y}^\top$:

$$\sigma_{min}(\mathbf{Y}\mathbf{Y}^\top) \leqslant \sigma_{min}(\mathbf{Y}\mathbf{N}^{(0)}\mathbf{N}^{(1)}\mathbf{Y}^\top) \leqslant \sigma_{max}(\mathbf{Y}\mathbf{N}^{(0)}\mathbf{N}^{(1)}\mathbf{Y}^\top) \leqslant \sigma_{max}(\mathbf{Y}\mathbf{Y}^\top) \qquad (46)$$

where inequalities become equalities, only if there is a right singular vector $\boldsymbol{\alpha}$ of $\mathbf{Y}^\top$ that is orthogonal to both $\mathbf{e}^{(0)}$ and $\mathbf{e}^{(1)}$.

**Assumption 7: None of the right singular vector of $\mathbf{Y}^\top$ is orthogonal to both $\mathbf{e}^{(0)}$ and $\mathbf{e}^{(1)}$**

Based on Assumption 7 and result in equation 46, following holds:

$$\sigma_{min}(\frac{\mathbf{Y}\mathbf{Y}^\top}{N}) < \sigma_{min}(\frac{\mathbf{Y}\mathbf{N}^{(0)}\mathbf{N}^{(1)}\mathbf{Y}^\top}{N}) < \sigma_{max}(\frac{\mathbf{Y}\mathbf{N}^{(0)}\mathbf{N}^{(1)}\mathbf{Y}^\top}{N}) < \sigma_{max}(\frac{\mathbf{Y}\mathbf{Y}^\top}{N}) \qquad (47)$$

Finally, for small enough $\delta_1$ and large enough $N$ with probability $1 - \epsilon$ with $\epsilon < \frac{2}{e}$, the following inequalities can be written:

$$\sigma_{min}(\frac{1}{N}\mathbf{Z}\mathbf{Z}^\top) \leqslant \sigma_{min}(\frac{\mathbf{Y}\mathbf{Y}^\top}{N}) + \delta_1 + \mathcal{O}\left(\frac{log(2/\epsilon)}{N}\right)$$

$$< \sigma_{min}(\frac{\mathbf{Y}\mathbf{N}^{(0)}\mathbf{N}^{(1)}\mathbf{Y}^\top}{N}) - \mathcal{O}\left(\frac{log(2/\epsilon)}{N}\right)$$

$$\leqslant \sigma_{min}(\frac{1}{N}\mathbf{Z}^{MShift}(\mathbf{Z}^{MShift})^\top)$$

$$\leqslant \sigma_{max}(\frac{1}{N}\mathbf{Z}^{MShift}(\mathbf{Z}^{MShift})^\top)$$
$$(48)$$

$$\leqslant \sigma_{max}(\frac{\mathbf{Y}\mathbf{N}^{(0)}\mathbf{N}^{(1)}\mathbf{Y}^\top}{N}) + \delta_1 + \mathcal{O}\left(\frac{log(2/\epsilon)}{N}\right)$$

$$< \sigma_{max}(\frac{\mathbf{Y}\mathbf{Y}^\top}{N}) - \mathcal{O}\left(\frac{log(2/\epsilon)}{N}\right)$$

$$\leqslant \sigma_{max}(\frac{1}{N}\mathbf{Z}\mathbf{Z}^\top)$$

Therefore, equation 48 proves that $1 - \left(\frac{\sigma_{min}(\mathbf{Z}\mathbf{Z}^\top)}{\sigma_{max}(\mathbf{Z}\mathbf{Z}^\top)}\right) > 1 - \left(\frac{\sigma_{min}(\mathbf{Z}^{MShift}(\mathbf{Z}^{MShift})^\top)}{\sigma_{max}(\mathbf{Z}^{MShift}(\mathbf{Z}^{MShift})^\top)}\right)$ with probability $1 - \epsilon$. Unifying this result with Equations equation 35 and equation 36 concludes that shifted model by matrices $\mathbf{N}^{(0)}\mathbf{N}^{(1)}$ converges faster compared to the vanilla model with high probability in the considered learning environment for node classification.

## C   Proof of Theorem 2

Define the sample mean after normalization layer by $\bar{\boldsymbol{\mu}}^{(n)} \in \mathbb{R}^F, n = 0, 1$. Then,

$$\|\bar{\boldsymbol{\mu}}^{(0)} - \bar{\boldsymbol{\mu}}^{(1)}\| = \|\frac{1}{|\mathcal{S}^0|} \sum_{j \in \mathcal{S}^0} \bar{\mathbf{h}}_j^{(0)} - \frac{1}{|\mathcal{S}^1|} \sum_{j \in \mathcal{S}^1} \bar{\mathbf{h}}_j^{(1)}\|$$

$$\|\boldsymbol{\mu}^{(0)} - \boldsymbol{\mu}^{(1)}\| = \|\frac{1}{|\mathcal{S}^0|} \sum_{j \in \mathcal{S}^0} \text{Act}(\bar{\mathbf{h}}_j^{(0)}) - \frac{1}{|\mathcal{S}^1|} \sum_{j \in \mathcal{S}^1} \text{Act}(\bar{\mathbf{h}}_j^{(1)})\|_1. \tag{49}$$

We can write $\bar{\mathbf{h}}_j^{(0)} = \bar{\boldsymbol{\mu}}^{(0)} + \bar{\boldsymbol{\delta}}_j^{(0)}, \forall j = 1 \cdots |\mathcal{S}^0|$ and $\bar{\mathbf{h}}_j^{(1)} = \bar{\boldsymbol{\mu}}^{(1)} + \bar{\boldsymbol{\delta}}_j^{(1)}, \forall j = 1 \cdots |\mathcal{S}^1|$. If the activation function $\text{Act}()$ is Lipschitz continuous with Lipschitz constant $L$ (applies to several nonlinear activations, such as rectified linear unit (ReLU), sigmoid), the following holds:

$$\text{Act}(\bar{\mu}_i^{(0)}) - L|\bar{\delta}_{i,j}^{(0)}| \leqslant \text{Act}(\bar{h}_{i,j}^{(0)}) = \text{Act}(\bar{\mu}_i^{(0)} + \bar{\delta}_{i,j}^{(0)})$$

$$\leqslant \text{Act}(\bar{\mu}_i^{(0)}) + L|\bar{\delta}_{i,j}^{(0)}|, \forall i = 1, \cdots F$$

$$\text{Act}(\bar{\boldsymbol{\mu}}^{(0)}) - L|\bar{\boldsymbol{\delta}}_j^{(0)}| \leqslant \text{Act}(\bar{\mathbf{h}}_j^{(0)}) = \text{Act}(\bar{\boldsymbol{\mu}}^{(0)} + \bar{\boldsymbol{\delta}}_j^{(0)})$$

$$\leqslant \text{Act}(\bar{\boldsymbol{\mu}}^{(0)}) + L|\bar{\boldsymbol{\delta}}_j^{(0)}|, \forall j = 1, \cdots |\mathcal{S}^0| \tag{50}$$

where $|.|$ takes the element-wise absolute value of the input. The same inequalities can also be written for $\mathcal{S}^1$:

$$\text{Act}(\bar{\boldsymbol{\mu}}^{(1)}) - L|\bar{\boldsymbol{\delta}}_j^{(1)}| \leqslant \text{Act}(\bar{\mathbf{h}}_j^{(1)}) = \text{Act}(\bar{\boldsymbol{\mu}}^{(1)} + \bar{\boldsymbol{\delta}}_j^{(1)}) \leqslant \text{Act}(\bar{\boldsymbol{\mu}}^{(1)}) + L|\bar{\boldsymbol{\delta}}_j^{(1)}|,$$

$$\forall j = 1, \cdots |\mathcal{S}^1| \tag{51}$$

Based on Equations equation 49, equation 50, and equation 51, following holds:

$$\frac{1}{|\mathcal{S}^0|} \sum_{j \in \mathcal{S}^0} \left( \text{Act}(\bar{\boldsymbol{\mu}}^{(0)}) - L|\bar{\boldsymbol{\delta}}_j^{(0)}| \right) - \frac{1}{|\mathcal{S}^1|} \sum_{j \in \mathcal{S}^1} \left( \text{Act}(\bar{\boldsymbol{\mu}}^{(1)}) + L|\bar{\boldsymbol{\delta}}_j^{(1)}| \right) \leqslant \boldsymbol{\mu}^{(0)} - \boldsymbol{\mu}^{(1)}$$

$$\leqslant \frac{1}{|\mathcal{S}^0|} \sum_{j \in \mathcal{S}^0} \left( \text{Act}(\bar{\boldsymbol{\mu}}^{(0)}) + L|\bar{\boldsymbol{\delta}}_j^{(0)}| \right) - \frac{1}{|\mathcal{S}^1|} \sum_{j \in \mathcal{S}^1} \left( \text{Act}(\bar{\boldsymbol{\mu}}^1) - L|\bar{\boldsymbol{\delta}}_j^{(1)}| \right) \tag{52}$$

$$\text{Act}(\bar{\boldsymbol{\mu}}^{(0)}) - \text{Act}(\bar{\boldsymbol{\mu}}^{(1)}) - \frac{1}{|\mathcal{S}^0|} \sum_{j \in \mathcal{S}^0} L|\bar{\boldsymbol{\delta}}_j^{(0)}| - \frac{1}{|\mathcal{S}^1|} \sum_{j \in \mathcal{S}^1} L|\bar{\boldsymbol{\delta}}_j^{(1)}| \leqslant \boldsymbol{\mu}^{(0)} - \boldsymbol{\mu}^{(1)}$$

$$\leqslant \text{Act}(\bar{\boldsymbol{\mu}}^{(0)}) - \text{Act}(\bar{\boldsymbol{\mu}}^{(1)}) + \frac{1}{|\mathcal{S}^0|} \sum_{j \in \mathcal{S}^0} L|\bar{\boldsymbol{\delta}}_j^{(0)}| + \frac{1}{|\mathcal{S}^1|} \sum_{j \in \mathcal{S}^1} L|\bar{\boldsymbol{\delta}}_j^{(1)}| \tag{53}$$

Define $\mathbf{a} := \text{Act}(\bar{\boldsymbol{\mu}}^{(0)}) - \text{Act}(\bar{\boldsymbol{\mu}}^{(1)}) - \frac{1}{|\mathcal{S}^0|} \sum_{j \in \mathcal{S}^0} L|\bar{\boldsymbol{\delta}}_j^{(0)}| - \frac{1}{|\mathcal{S}^1|} \sum_{j \in \mathcal{S}^1} L|\bar{\boldsymbol{\delta}}_j^{(1)}|$ and $\mathbf{b} := \text{Act}(\bar{\boldsymbol{\mu}}^{(0)}) - \text{Act}(\bar{\boldsymbol{\mu}}^{(1)}) + \frac{1}{|\mathcal{S}^0|} \sum_{j \in \mathcal{S}^0} L|\bar{\boldsymbol{\delta}}_j^{(0)}| + \frac{1}{|\mathcal{S}^1|} \sum_{j \in \mathcal{S}^1} L|\bar{\boldsymbol{\delta}}_j^{(1)}|$. Equation equation 53 leads to:

$$|\mu_i^{(0)} - \mu_i^{(1)}| \leqslant \max(|a_i|, |b_i|), \forall i = 1, \cdots, F. \tag{54}$$

If we consider the case, $|a_i| \geqslant |b_i|$. Then:

$$|\mu_i^{(0)} - \mu_i^{(1)}| \leqslant \left| \text{Act}(\bar{\mu}_i^{(0)}) - \text{Act}(\bar{\mu}_i^{(1)}) - \frac{1}{|\mathcal{S}^0|} \sum_{j \in \mathcal{S}^0} L|\bar{\delta}_{j,i}^{(0)}| - \frac{1}{|\mathcal{S}^1|} \sum_{j \in \mathcal{S}^1} L|\delta_{j,i}^{(1)}| \right|$$

$$\leqslant \left| \text{Act}(\bar{\mu}_i^{(0)}) - \text{Act}(\bar{\mu}_i^{(1)}) \right| + \left| \frac{1}{|\mathcal{S}^0|} \sum_{j \in \mathcal{S}^0} L|\bar{\delta}_{j,i}^{(0)}| \right| + \left| \frac{1}{|\mathcal{S}^1|} \sum_{j \in \mathcal{S}^1} L|\bar{\delta}_{j,i}^{(1)}| \right| \tag{55}$$

$$\leqslant \left| \text{Act}(\bar{\mu}_i^{(0)}) - \text{Act}(\bar{\mu}_i^{(1)}) \right| + L \left| \bar{\Delta}_i^{(0)} \right| + L \left| \bar{\Delta}_i^{(1)} \right|,$$

where $\bar{\Delta}_i^{(0)} := \max_j |\bar{\delta}_{j,i}^{(0)}|$ and $\bar{\Delta}_i^{(1)} := \max_j |\bar{\delta}_{j,i}^{(1)}|$.

Consider the term $\text{Act}(\bar{\mu}_i^{(0)}) - \text{Act}(\bar{\mu}_i^{(1)})$:

$$\text{Act}(\bar{\mu}_i^{(0)}) - \text{Act}(\bar{\mu}_i^{(1)}) = \text{Act}(\bar{\mu}_i^{(0)} + \bar{\mu}_i^{(1)} - \bar{\mu}_i^{(1)}) - \text{Act}(\bar{\mu}_i^{(1)}). \tag{56}$$

Utilizing Equations equation 50 and equation 51, $\text{Act}(\bar{\mu}_i^{(0)} + \bar{\mu}_i^{(1)} - \bar{\mu}_i^{(1)}) - \text{Act}(\bar{\mu}_i^{(1)})$ can be bounded by below and above:

$$\begin{aligned}
\text{Act}(\bar{\mu}_i^{(1)}) - L|\bar{\mu}_i^{(0)} - \bar{\mu}_i^{(1)}| - \text{Act}(\bar{\mu}_i^{(1)}) &\leqslant \text{Act}(\bar{\mu}_i^{(0)} + \bar{\mu}_i^{(1)} - \bar{\mu}_i^{(1)}) - \text{Act}(\bar{\mu}_i^{(1)}) \\
&\leqslant \text{Act}(\bar{\mu}_i^{(1)}) + L|\bar{\mu}_i^{(0)} - \bar{\mu}_i^{(1)}| - \text{Act}(\bar{\mu}_i^{(1)})
\end{aligned} \tag{57}$$

$$-L|\bar{\mu}_i^{(0)} - \bar{\mu}_i^{(1)}| \leqslant \text{Act}(\bar{\mu}_i^{(0)} + \bar{\mu}_i^{(1)} - \bar{\mu}_i^{(1)}) - \text{Act}(\bar{\mu}_i^{(1)}) \leqslant L|\bar{\mu}_i^{(0)} - \bar{\mu}_i^{(1)}| \tag{58}$$

$$\left| \text{Act}(\bar{\mu}_i^{(0)}) - \text{Act}(\bar{\mu}_i^{(1)}) \right| \leqslant L|\bar{\mu}_i^{(0)} - \bar{\mu}_i^{(1)}| \tag{59}$$

Therefore:

$$|\mu_i^{(0)} - \mu_i^{(1)}| \leqslant L|\bar{\mu}_i^{(0)} - \bar{\mu}_i^{(1)}| + L\left|\bar{\Delta}_i^{(0)}\right| + L\left|\bar{\Delta}_i^{(1)}\right|, \forall i \text{ such that } |a_i| \geqslant |b_i|. \tag{60}$$

Next step is to consider the case, $|a_i| < |b_i|$. For this case, following inequalities hold:

$$\begin{aligned}
|\mu_i^{(0)} - \mu_i^{(1)}| &\leqslant \left| \text{Act}(\bar{\mu}_i^{(0)}) - \text{Act}(\bar{\mu}_i^{(1)}) + \frac{1}{|\mathcal{S}^0|}\sum_{j\in\mathcal{S}^0} L|\bar{\delta}_{j,i}^{(0)}| + \frac{1}{|\mathcal{S}^1|}\sum_{j\in\mathcal{S}^1} L|\bar{\delta}_{j,i}^{(1)}| \right| \\
&\leqslant \left| \text{Act}(\bar{\mu}_i^{(0)}) - \text{Act}(\bar{\mu}_i^{(1)}) \right| + \left| \frac{1}{|\mathcal{S}^0|}\sum_{j\in\mathcal{S}^0} L|\bar{\delta}_{j,i}^{(0)}| \right| + \left| \frac{1}{|\mathcal{S}^1|}\sum_{j\in\mathcal{S}^1} L|\bar{\delta}_{j,i}^{(1)}| \right| \\
&\leqslant \left| \text{Act}(\bar{\mu}_i^{(0)}) - \text{Act}(\bar{\mu}_i^{(1)}) \right| + L\left|\bar{\Delta}_i^{(0)}\right| + L\left|\bar{\Delta}_i^{(1)}\right| \\
&\leqslant L|\bar{\mu}_i^{(0)} - \bar{\mu}_i^{(1)}| + L\left|\bar{\Delta}_i^{(0)}\right| + L\left|\bar{\Delta}_i^{(1)}\right|, \forall i \text{ such that } |a_i| \geqslant |b_i|.
\end{aligned} \tag{61}$$

Combining Equations equation 60 and equation 61, the following inequality can be written:

$$|\mu_i^{(0)} - \mu_i^{(1)}| \leqslant L|\bar{\mu}_i^{(0)} - \bar{\mu}_i^{(1)}| + L\left|\bar{\Delta}_i^{(0)}\right| + L\left|\bar{\Delta}_i^{(1)}\right|, \forall i = 1, \ldots, F. \tag{62}$$

which concludes:

$$\|\boldsymbol{\mu}^{(0)} - \boldsymbol{\mu}^{(1)}\| \leqslant L\left( \|\bar{\boldsymbol{\mu}}^{(0)} - \bar{\boldsymbol{\mu}}^{(1)}\| + \|\bar{\boldsymbol{\Delta}}^{(0)}\| + \|\bar{\boldsymbol{\Delta}}^{(1)}\| \right). \tag{63}$$

## D  Hyperparameters

We provide the selected hyperparameter values for the GNN model and the proposed framework for the reproducibility of the presented results. In the GNN-based classifier, weights are initialized utilizing Glorot initialization (Glorot & Bengio, 2010). All models are trained for 1000 epochs by employing Adam optimizer (Kingma & Ba, 2014) together with a learning rate of $10^{-3}$ and $\ell_2$ weight decay factor of $10^{-5}$. A 2-layer GCN network followed by a linear layer is employed for node classification. Hidden dimension of the node representations is selected as 64 on all datasets.

The results for baseline schemes, covariance, adversarial, $HTR_{DDP}$ regularizers, and FairGNN (Dai & Wang, 2021) are obtained by choosing corresponding hyperparameters (the multiplying factors for these regularizers in the overall loss) via grid search on cross-validation sets with 5 different data splits. Specifically, a grid search on the values $10^9, 10^{10}, 10^{11}$ is executed for covariance-based regularizer. Furthermore, the values $0.01, 0.1, 1.10$ are examined as the multiplier for adversarial regularizer. FairGNN employs both covariance-based and adversarial regularizers, therefore its parameter selection is the unification of the previous two

hyperparameter selection methods. Finally, the parameters $0.01, 0.1, 1, 10$ are examined for the $HTR_{DDP}$ regularizer.

Hyperparameter values for the proposed FairNorm whose results are presented in Section 5.2 can be found in Table 6. Note that the candidate hyperparameter values for $\tau$ and $\kappa$ are selected to balance the significance of $\mathcal{L}_c$, $\mathcal{L}_\mu$, and $\mathcal{L}_\Delta$ in equation 8 in terms of order of magnitude, so that each component has non-negligible yet non-dominant effect. These parameters are selected via grid search on cross-validation sets over 5 different data splits. The range of the parameters are selected based on the corresponding loss terms. Specifically, the values $10, 100, 1000$ values are examined for $\kappa$ in Pokec networks, and the values $0.01, 0.1, 1$ values are examined for $\kappa$ in Recidivism network. After the selection of best $\kappa$ value, the $\tau$ is selected together with the fixed, best $\kappa$ value. While the values $10^{-7}, 10^{-8}, 10^{-9}, 10^{-10}$ values are examined for $\tau$ in Pokec networks, the values $10^{-8}, 10^{-9}, 10^{-10}, 10^{-11}$ values are examined for $\kappa$ in Recidivism network

Table 6: Utilized $\kappa$ and $\tau$ values for the presented results in Table 2

| ReLU | Pokec-z | Pokec-n | Recidivism |
|---|---|---|---|
| $\kappa/\tau$ | $100/10^{-7}$ | $1000/10^{-9}$ | $0.01/10^{-10}$ |
| Sigmoid | Pokec-z | Pokec-n | Recidivism |
| $\kappa/\tau$ | $10/10^{-7}$ | $100/10^{-8}$ | $0.01/10^{-10}$ |

## E  Experimental Results for GraphSAGE

In order to demonstrate the flexibility of the proposed scheme for different GNN models, we provide further experimental results herein for GraphSAGE operators proposed in Hamilton et al. (2017b) together with mean aggregation. The obtained results for node classification on Pokec networks and the Recidivism graph with ReLU activation are presented in Table 7. The results in Table 7 confirm the efficacy of FairNorm also for the GraphSAGE-based GNN, since FairNorm still achieves better or similar fairness measures compared to other fairness-aware baselines together with similar utility. Note that the hyperparameters for this set of experiments are selected in the same way that is described in Appendix D.

Table 7: Comparative Results for GraphSAGE with ReLU activation

| | Pokec-z | | | Pokec-n | | | Recidivism | | |
|---|---|---|---|---|---|---|---|---|---|
| | Acc (%) | $\Delta_{SP}$ (%) | $\Delta_{EO}$ (%) | Acc (%) | $\Delta_{SP}$ (%) | $\Delta_{EO}(\%)$ | Acc (%) | $\Delta_{SP}$ (%) | $\Delta_{EO}(\%)$ |
| NoNorm | $\mathbf{70.16} \pm 0.8$ | $6.37 \pm 1.2$ | $5.32 \pm 1.1$ | $68.73 \pm 0.6$ | $2.69 \pm 1.3$ | $1.76 \pm 1.6$ | $97.93 \pm 0.1$ | $9.37 \pm 0.9$ | $1.42 \pm 0.1$ |
| M-Norm | $69.94 \pm 0.5$ | $2.82 \pm 1.7$ | $3.52 \pm 1.6$ | $68.96 \pm 0.9$ | $2.45 \pm 1.8$ | $1.82 \pm 1.5$ | $\mathbf{99.01} \pm 0.1$ | $9.26 \pm 0.7$ | $\mathbf{0.56} \pm 0.2$ |
| Covariance | $69.95 \pm 0.7$ | $3.30 \pm 2.0$ | $3.41 \pm 2.2$ | $\mathbf{69.61} \pm 0.6$ | $3.26 \pm 1.7$ | $2.16 \pm 1.4$ | $98.68 \pm 0.2$ | $9.21 \pm 0.9$ | $0.74 \pm 0.5$ |
| Adversarial | $68.99 \pm 1.0$ | $1.94 \pm 1.6$ | $\mathbf{1.97} \pm 1.6$ | $68.94 \pm 1.0$ | $3.24 \pm 1.9$ | $2.43 \pm 2.0$ | $98.88 \pm 0.1$ | $9.13 \pm 0.8$ | $0.73 \pm 0.6$ |
| FairGNN | $68.92 \pm 1.1$ | $2.79 \pm 1.4$ | $2.42 \pm 1.5$ | $68.99 \pm 0.9$ | $2.96 \pm 2.3$ | $2.12 \pm 1.4$ | $98.82 \pm 0.1$ | $9.36 \pm 0.6$ | $0.66 \pm 0.5$ |
| $HTR_{DDP}$ | $\mathbf{70.16} \pm 0.5$ | $2.51 \pm 1.7$ | $2.90 \pm 1.8$ | $69.35 \pm 1.9$ | $2.94 \pm 2.0$ | $1.66 \pm 1.2$ | $98.98 \pm 0.1$ | $9.36 \pm 0.7$ | $\mathbf{0.56} \pm 0.1$ |
| FairNorm | $69.50 \pm 0.8$ | $\mathbf{1.37} \pm 1.5$ | $\mathbf{2.00} \pm 1.9$ | $69.05 \pm 1.1$ | $\mathbf{2.38} \pm 1.3$ | $\mathbf{1.30} \pm 0.6$ | $98.98 \pm 0.1$ | $\mathbf{9.01} \pm 0.8$ | $\mathbf{0.55} \pm 0.6$ |

## F  Validation Accuracy Curves

Figure 1 has already shown that FairNorm can achieve faster convergence in training compared to the framework where no normalization is employed. Figure 2 is provided herein to demonstrate the validation accuracy over 1000 epochs for the case where normalization is not employed, GraphNorm Cai et al. (2021), and our framework FairNorm. Note that models that achieve the best validation set accuracy are utilized to obtain the test set performances listed in Table 2. The curves in Figure 2 further signify that FairNorm and GraphNorm can converge to the model with the best validation set accuracy faster compared to the scheme that does not use normalization.

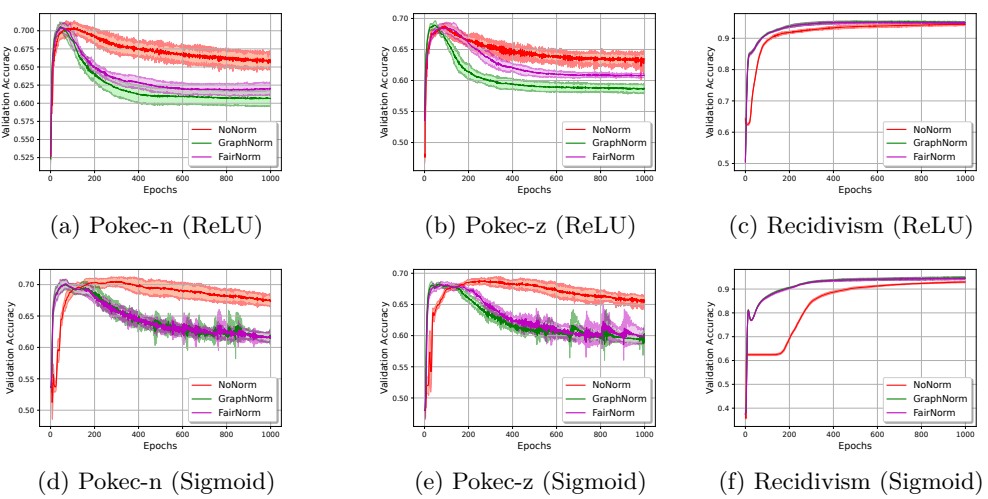

(a) Pokec-n (ReLU)  (b) Pokec-z (ReLU)  (c) Recidivism (ReLU)

(d) Pokec-n (Sigmoid)  (e) Pokec-z (Sigmoid)  (f) Recidivism (Sigmoid)

Figure 2: Validation curves for different graph data sets when the normalization is not applied (Nonorm) and applied with/without fairness consideration (FairNorm/GraphNorm).

