# OpenReview forum: "Fast&Fair: Training Acceleration and Bias Mitigation for GNNs"
_TMLR — Accepted by TMLR_

### Review · Reviewer_YEuY · 2023-01-15

**Summary Of Contributions:**

This paper proposes FairNorm for reducing the bias in GNN-based learning while simultaneously providing faster convergence. FairNorm addresses to normalize each different sensitive group individually and introduces fairness regularizes on the learnable parameters of normalization layers to reduce the bias in GNNs. The technical properties of FairNorm for improving the convergence and reducing the bias are derived, based on the previous research progress in normalization and fairness learning of GNNs. Experimental results also empirically support the effectiveness of the proposed method for improving fairness and convergence speed of GNNs.

**Audience:**

Yes

**Broader Impact Concerns:**

No impact concerns.

**Claims And Evidence:**

Yes

**Requested Changes:**

1.More discussion to related works
2.Better to re-organize/reword Lemma 1 and Theorem 1, in case of over-much overlap in description, compared to Cai et al (2021)

**Strengths And Weaknesses:**

**Strengths:**
1. The motivation of this paper is very clear. Based on the previous research progresses that normalization can improve the convergence and stability of GNNs, this paper further addresses the fairness scenarios of GNNs, and designs FairNorm to improve the training stability and fairness of GNNs. It is clear that the proposed method can obtain good performance.
2. This paper is well written and I am glad to read it.

**Weaknesses:**

1. The techniqual contribution of this paper is somewhat incremental. This paper is a typic A+B paper，where A is that normalization can accelerate the convergence (and preventing the over-smooth problem of GNN) and B is that the fairness-aware regularization can improve the fairness of the GNN. This paper thus proposes the framework that combining normalization and fairness.  I recognize that this paper extends Cai et al (2021) by using group-wise normalization, and extends Li et al (2020); Kose&Shen (2022) by considering the normalization operation. However, the following points make me feel the contributions are incremental and should be further amplified:
(1)	The Lemma 1 and Theorem 1 is a straightforward extension from Li et al 2020. Furthermore, when I compare the corresponding content of Li et al (2020) and this paper, I find over-much overlap in description between them. I think it is better to re-organize/reword (even just only cite Li et al (2020)) this part (Lemma 1 and Theorem 1).
(2)	The main extension of this paper over Li et al (2020) is that it normalizes different sensitive attributions independently. The similar idea using normalization has been proposed extensively in previous work for different scenarios (See weakness 2), I think this paper should discuss these papers in “related work” section.

2. This paper misses some closely related papers [1][2][3][4][5]. In terms of the normalization for GNN, there exists at least two related work [1]and [2]; In terms of normalizing over different groups (e.g, domains), there are a bunch of papers, e.g., for domain adaptation[3], for adversarial robust[4], for image inpainting[5]. I suggest the authors refer to the normalization survey paper [6] for an overall picture.

3. I think section 3.1 is somewhat redundant to the “Normalization” paragraph in Section 2. I believe section 3.1 should provide the technical details of the main normalization for GNNs (including the mathematical notation and main formulation).

Other minor:  What is $r_{i,j}^{(n)}$ in page 6?


Citation:
[1]Towards Deeper Graph Neural Networks with Differentiable Group Normalization, Neurips 2020
[2]PairNorm: Tackling Oversmoothing in GNNs. ICLR, 2020
[3] Domain-Specific Batch Normalization for Unsupervised Domain Adaptation. CVPR, 2019
[4] Towards Defending Multiple Adversarial Perturbations via Gated Batch Normalization. arXiv:2012.01654, 2020
[5] Region Normalization for Image Inpainting. AAAI, 2020
[6] Normalization Techniques in Training DNNs: Methodology, Analysis and Application. arXiv:2009.12836, 2020.

---

### Review · Reviewer_jZmj · 2023-02-19

**Summary Of Contributions:**

This paper aims at the bias problem in Graph Neural Networks (GNNs) and introduce a group-wise normalization to improve the fairness. Specifically, the authors propose to apply normalizations to different sensitive groups individually and include shift operations in groups to improve convergence speed. To further improve fairness, the authors introduce fairness-aware regularizers via the upper bound of difference between mean representations of different sensitive groups. Additionally, the authors provide theoretical analysis on the convergence and regularizers.

**Audience:**

Yes

**Broader Impact Concerns:**

No concerns.

**Claims And Evidence:**

Yes

**Requested Changes:**

Questions:

1. Please discuss the relationship between graph norm and fair norm in Theorem 1.
2. Please provide more results on different GNN models.

Minors：
1. Some definitions of notations are missing, e.g., F in the feature matrices.
2. Add bold to highlight the best results and settings in Table 3 and Table 4.

Missing reference:

There exists relevant works which discuss the normalization layers and stability.

[1]. Understanding and Improving Layer Normalization.

[2]. Towards Stable and Robust AdderNets.

[3]. Delving Into the Estimation Shift of Batch Normalization in a Network.

[4]. Learning graph normalization for graph neural networks.


**Strengths And Weaknesses:**

Strengths:

* The paper is well-organized and well-written.
* Sufficient theoretical analysis based on proposed method
* The evaluation is conducted on various datasets, and the proposed method shows better performance than other baselines.

Weakness:

* The theoretical analysis in Theorem 1 is similar to the one in [a]. Since the shift operation in GNNs is proposed in [a] which leads to the convergence advantage, it would be better for the authors to compare or discuss the relationship between group-wise shifts and the one in [a] instead of the one without shift in Theorem 1.
* The proposed method is evaluated on a tiny GCN model. It would be better for the authors to provide more evaluation results on different GNN models.


[a]. GraphNorm: A Principled Approach to Accelerating Graph Neural Network Training. ICML 2021.

---

### Review · Reviewer_5jyf · 2023-03-11

**Summary Of Contributions:**

This work proposes FairNorm, which introduces individual normalization operators and fairness regularizers into their framework for GNN-based graph learning. It aims to reduce bias and improve the speed of convergence. Experiments are performed on node classification tasks to empirically evaluate the proposed method.

**Audience:**

Yes

**Broader Impact Concerns:**

No concern on the ethical implications.

**Claims And Evidence:**

Yes

**Requested Changes:**

Please fix issues mentioned in “weaknesses”.

**Strengths And Weaknesses:**

Strengths:
- The group-wise normalization over different sensitive groups is novel and seems promising.
- The authors provide solid theoretical analyses to support their proposal.
- Remark 1 and 2 shows the flexibility and applicability of their regularizers.

Weaknesses:
- Fig. 1 only show the training accuracy curves. It would be better to also include validation/test curves to show there is no overfitting.
- In Tab. 3 and 4, $\tau$ is set to very small values (1e-7 to 1e-9), and adjusting it has marginal impacts. Could the authors explain why does $L_\Delta$ has such small weights? And is it really significant to the result?
- In Tab. 3 and 4, the value $\kappa$ varies a lot when using different network. This is also a minor drawback, because when generalizing the proposed method to other networks, a lot of fine-tuning of hyper-parameters may be demanded.

---

### Review · Reviewer_njwj · 2023-03-12

**Summary Of Contributions:**

A new framework is proposed to reduce bias while providing a higher convergence speed in the GNN-based learning scheme. To achieve the fairness guarantee based on the normalization framework, a fairness-aware regularizer is introduced on the trainable parameters of the normalization layers. Empirical results are provided.

**Audience:**

Yes

**Broader Impact Concerns:**

No concern on the ethical implications.

**Claims And Evidence:**

Yes

**Requested Changes:**

Please address the following questions:
1. In the objective function (8), why do you introduce two trade-off parameters with respect to $\mathcal{L}_{\mu}$ and $\mathcal{L}_{\Delta}$ respectively? Is there any theoretical reason? I am curious about this formulation because according to the results in Theorem 2, the mean bias term and the deviation term seem to have the same importance for bounding the distance between sample means of two groups.
2. As mentioned at the beginning of Section 4.1,  Li et al. (2020); Kose & Shen (2022) demonstrate that decreasing $\|\mu^{0}-\mu^{1}\|$ and $\Delta$ can effectively reduce bias in GNN-based learning, the results in the paper only analyze the term about mean sample difference ($\|\mu^{0}-\mu^{1}\|$). Then what about $\Delta$? How can we reduce $\Delta$ in the framework? Will reducing $\|\mu^{0}-\mu^{1}\|$ cause a smaller $\Delta$?

**Strengths And Weaknesses:**

Strengths: The idea of the paper is clear and novel. Based on the normalization structure which improves the convergence and stability of GNNs, this paper further introduces the fairness concern into the problem. The paper is well-organized.
Weaknesses: Although the author claimed that the proposed framework is the first attempt to improve fairness and convergence speed in a unified framework, the work seems to simply combine normalization and fairness together. The regularizer is not related to any well-known fairness metrics, e.g. demographic parity or equalized opportunity. In my opinion, proposing a new fairness measure and using it as a regularizer in the objective function is not novel enough without demonstrating the relationship between this measure and other well-known fairness metrics.

---

### Decision · Action_Editors · 2023-04-24

**Recommendation:** Accept as is

**Comment:**

This paper proposes a novel method called FairNorm, which addresses the bias problem in GNN-based learning and simultaneously provides faster convergence. FairNorm introduces group-wise normalization to normalize each different sensitive group individually and proposes fairness-aware regularizers via the upper bound of the difference between mean representations of different sensitive groups.  The authors have addressed all the reviewer's concerns during the review process.

**Audience:**

The findings of this paper could be of interest to individuals working in the fields related to GNN-based learning, normalization, and fairness. The paper proposes a novel method for improving fairness and convergence speed in GNNs and provides solid theoretical analyses and experimental results to support the effectiveness of the proposed method. Additionally, the paper addresses the bias problem in GNN-based learning, which is an important issue in the field of machine learning. Therefore, the findings of this paper could be of interest to researchers and practitioners in these fields.





**Claims And Evidence:**

The claims made in the submission are supported by accurate, convincing, and clear evidence. The paper proposes FairNorm, a novel method for improving the fairness and convergence speed of GNNs. The authors introduce group-wise normalization to address the bias problem in GNN-based learning and propose fairness-aware regularizers via the upper bound of the difference between mean representations of different sensitive groups. The authors provide solid theoretical analyses to support their proposal and demonstrate the flexibility and applicability of their regularizers. While the regularizer is not related to any well-known fairness metrics, the group-wise normalization over different sensitive groups is novel and promising. The experimental results also support the effectiveness of the proposed method in improving fairness and convergence speed of GNNs. Overall, the paper is well-organized and well-written.